# Regulation of the one carbon folate cycle as a shared metabolic signature of longevity

Andrea Annibal [1], Rebecca George Tharyan[1], Maribel Fides Schonewolff[1], Hannah Tam [1], Christian Latza[1], Markus Max Karl Auler[1], Sebastian Grönke[1], Linda Partridge[1] & Adam Antebi[1,2 ✉]

The metabolome represents a complex network of biological events that reflects the physiologic state of the organism in health and disease. Additionally, specific metabolites and metabolic signaling pathways have been shown to modulate animal ageing, but whether there are convergent mechanisms uniting these processes remains elusive. Here, we used high resolution mass spectrometry to obtain the metabolomic profiles of canonical longevity pathways in *C. elegans* to identify metabolites regulating life span. By leveraging the metabolomic profiles across pathways, we found that one carbon metabolism and the folate cycle are pervasively regulated in common. We observed similar changes in long-lived mouse models of reduced insulin/IGF signaling. Genetic manipulation of pathway enzymes and supplementation with one carbon metabolites in *C. elegans* reveal that regulation of the folate cycle represents a shared causal mechanism of longevity and proteoprotection. Such interventions impact the methionine cycle, and reveal methionine restriction as an underlying mechanism. This comparative approach reveals key metabolic nodes to enhance healthy ageing.

[1] Max Planck Institute for Biology of Ageing, Cologne, Germany. [2] Cologne Excellence Cluster on Cellular Stress Responses in Aging-Associated Diseases (CECAD), University of Cologne, Cologne, Germany. ✉email: aantebi@age.mpg.de

Cellular metabolism encompasses a highly integrated, complex network that supports the development, growth, and reproduction of the organism. Small molecule metabolites comprise basic building blocks for macromolecules and serve as essential carriers of energy and redox potential. Metabolites can also work as signaling molecules that regulate metabolic flux, epigenetic landscapes, gene regulatory networks as well as nutrient and growth signaling pathways[1]. Metabolic dysregulation contributes significantly to diseases such as cancer, cardiovascular disease, and inflammation, and manipulating metabolite levels in vivo can help restore metabolic balance and health[2].

More recently endogenous metabolites have also emerged as crucial modulators of animal longevity. These include various amino acids, alpha-ketoglutarate, spermidine, hexosamines, bile acids, nicotinamides, cannabinoids, ascarosides, and other natural compounds that regulate diverse aspects of signaling, metabolism, and homeostasis[3]. Moreover, many of the major longevity pathways are conserved regulators of metabolism, nutrient sensing, and growth[4]. Downregulation of insulin/IGF and mTOR signaling, reduced mitochondrial respiration, dietary restriction, and hormonal signals from the reproductive system can remodel metabolism, proteostasis, stress pathways, and immunity towards extended survival and longevity[5]. The question arises, do these diverse signaling pathways converge on shared metabolic outputs that are causal for an extended life? Here we show that canonical longevity pathways converge on the one-carbon folate cycle, and cause a decrease in levels of the intermediate 5 methyl tetrahydrofolate (5MTHF) and various methionine cycle intermediates. Genetic manipulation of pathways enzymes and supplementation experiments reveal that reduction of specific folate intermediates promotes longevity and proteoprotection as a common conserved mechanism that acts through methionine restriction.

## Results

### Metabolomic fingerprint of long-lived mutants.
To understand whether longevity pathways converge on common metabolic outputs, we performed high-resolution mass spectrometry on several long-lived mutant strains in *C. elegans*, to retrieve metabolic fingerprints. We used four canonical longevity mutants, namely insulin/IGF signaling (IIS)-deficient *daf-2(e1370)*, dietary restriction (DR) model *eat-2(ad465)*, mitochondrial respiration deficient *isp-1(qm150)* and germline-less *glp-1(e2141)ts* worms. Differentially regulated metabolites were characterized by mass spectrometry-based untargeted metabolomics, using reverse-phase liquid chromatography combined with electrospray ionization high-resolution accurate mass (ESI-HRAM) spectrometry. Using this method, we identified and quantified 145 unique metabolites representing different metabolic modules (Fig. 1a and Supplementary Fig. 1a–c, Supplementary Data 1). Partial least squares discriminant analysis (PLS-DA) of the biological replicates revealed a clear grouping according to genotype (Supplementary Fig. 1b), showing the high quality of the samples. In addition, both *glp-1* and WT grown at 25 °C as well as *eat-2* separated from the main cluster of *isp-1, daf-2*, and WT grown at 20 °C.

We first examined the metabolic signatures of each genotype individually. As expected, the relative levels of numerous metabolites were significantly changed (adj $p < 0.05$) in and *glp-1* (35), *daf-2* (59), *isp-1* (37), and *eat-2* (61), compared to wild-type (WT) (Fig. 1b and Supplementary Data 1). Some of these changes have been observed previously[6–11], confirming the validity of the approach, while others appeared novel (Supplementary Fig. 2 and Supplementary Table 2). The number of metabolites that were uniquely regulated in only one genotype

relative to WT was 13 in *daf-2*, 7 in *isp-1*,15 in *glp-1*, and 15 in *eat-2* (adj $p < 0.05$) (Fig. 1b and Supplementary Data 1).

As multiple individual metabolites were changed in each of the longevity pathways (Fig. 1a), we first sought to evaluate KEGG pathway enrichment for each genotype. We uploaded the significantly changed ($p$ adj < 0.05) metabolites to MetaboAnalyst (https://www.metaboanalyst.ca). This platform uses a reference metabolome to determine if the experimental metabolite set is overrepresented for certain KEGG-defined metabolic terms compared to random chance. Major enriched terms in *glp-1* vs WT-25 comparison included purine metabolism, phenylalanine, tyrosine and tryptophan biosynthesis, glutathione metabolism, glutamine, and glutamate metabolism, and branched-chain amino acids. *daf-2* vs WT comparisons revealed enrichment in purine, glutamine, and glutamate metabolism, branched-chain amino acids, pantothenate and CoA, and arginine biosynthesis. *isp-1* vs WT comparisons showed enrichment in glutamine and glutamate metabolism, arginine metabolism, pantothenate and CoA biosynthesis, purine metabolism, alanine, aspartate metabolism, and butanoate metabolism. *eat-2* vs WT comparisons showed enrichment for glutamine and glutamate metabolism, purine metabolism, arginine, nicotinamide, and branched-chain amino acid metabolism. (Supplementary Fig. 1d–g and Supplementary Table 3). Several KEGG terms were shared by all genotypes, including D-glutamine and D-glutamate metabolism, aminoacyl-tRNA biosynthesis, valine, leucine, and isoleucine biosynthesis, and purine metabolism, reflecting general regulation of amino acid and nucleotide metabolism in common. A caveat of enrichment analysis is that it is biased towards metabolites that can be readily measured on our platform, and might not highlight individual metabolites that are significantly changed but whose metabolic pathways are not necessarily enriched.

### Changed metabolites common across pathways.
We next leveraged changes in individual metabolites across all four mutants to determine if there were any common specific features. Although several metabolites showed trends in common across pathways, none emerged as significant (adj $p < 0.05$) from this analysis (Fig. 1b and Supplementary Data 1). Uracil, isoleucine, glutamine, lysine, appeared higher in all genotypes (Fig. 1a), but reached significance only in 2 or 3 backgrounds (Supplementary Fig. 2 and Supplementary Data 1). Erythronic acid, phospho-threonine, propionyl carnitine, kynurenine, gamma-glutamyl cysteine, methyl quinoline, glucosamine phosphate, phospho pantethine, FAD, thiamine, 2-amino benzoic acid, S-adenosyl methionine, dimethylarginine, appeared lower in all genotypes (Fig. 1a), but significant only in 1–3 backgrounds (Supplementary Fig. 2 and Supplementary Data 1).

*glp-1* mutants are sterile and lack germline, which could give rise to a disparate metabolic profile. We, therefore, limited our search to the metabolites that were commonly and significantly regulated in three non-sterile genotypes, *daf-2, eat-2,* and *isp-1* (adj $p < 0.05$, Supplementary Fig. 2 and Supplementary Data 1). From this comparison, significantly upregulated metabolites included several amino acids (isoleucine, leucine, glutamine, glutamic acid). Folate and methionine metabolism intermediates were downregulated (5 methyl tetrahydrofolate, S-adenosyl methionine, dimethylarginine), whereas homocysteine was increased. Nucleotides and related metabolites were also variously dysregulated (uracil, guanosine, inosine, cyclic GMP, NMN, FAD). Moreover, we found changes in phosphothreonine, propionyl carnitine, kynurenine, pantethine monophosphate, pantothenic acid, glucosamine phosphate, gamma-glutamyl cysteine, thiamine, and 2-amino benzoic acid (Supplementary Fig. 2 and Supplementary Data 1).

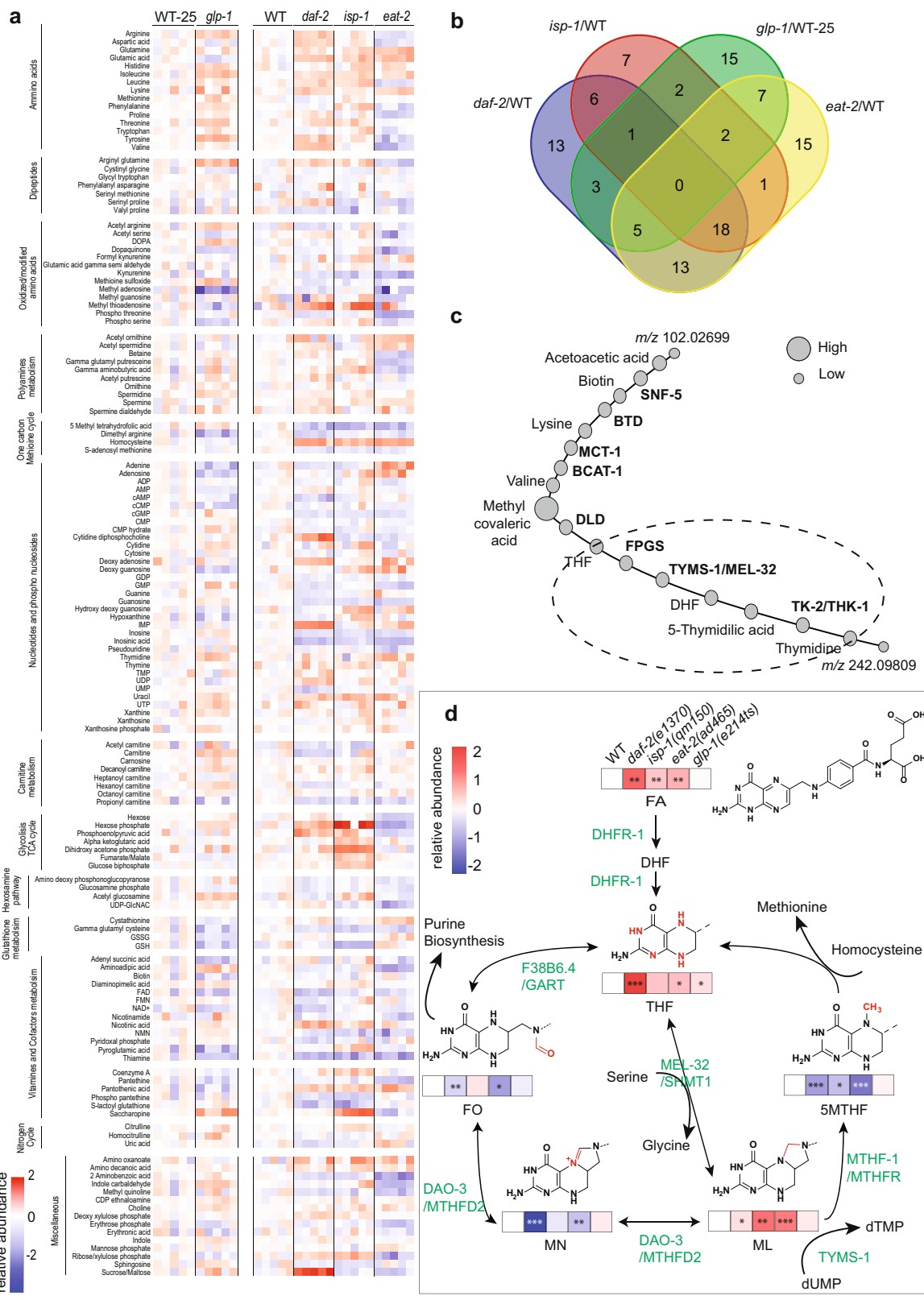

Taking advantage of our unbiased metabolomics acquisition, we additionally retrieved sixty unassigned $m/z$ values that were differentially regulated in all genotypes (Supplementary Table 4). These uncharacterized $m/z$ values were submitted to a pathway predictor software (PIUMet)[12]. The PIUMet algorithm identified 75% of the submitted features, correctly assigned the already reported metabolites, and pinpointed 10 new $m/z$ features. To find the connection between the assigned features and a common pathway, the algorithm linked the identified compounds together based on high-confidence protein–protein and protein–metabolites interactions (PPI) (Supplementary Table 5). The network revealed high-confidence links of organic acids, branched-chain amino acid, and folic acid metabolism (Fig. 1c and Supplementary Table 5). Given the biological importance of the folic acid pathway in human

**Fig. 1 Regulation of the folic acid cycle is altered in longevity mutants. a** Untargeted metabolomic analysis of *daf-2(e1370)*, *eat-2(ad465)*, *isp-1(qm150)*, and *glp-1(e2141)ts* worms at day 1 of adulthood. *glp-1(e2141)ts* worms are compared with wild-type (WT) control, which undergoes the same 25 °C thermal shift as the mutant. Heat map containing all biological replicates, indicates the relative abundance of metabolite concentrations relative to the wild-type average, including both significant and non-significant changes, (listed in Supplementary Data 1). Metabolites are manually grouped into different functional categories. **b** Venn diagram of the significantly changed metabolites (adj *p* < 0.05) for each genotype showing unique and overlapping compounds. **c** Metabolic-protein network of unknown and known features created by the PIUMet algorithm (http://fraenkel-nsf.csbi.mit.edu/piumet2/). The degree of confidence of the PPI algorithm is represented by node diameters. Additional parameters are found in Supplementary Table 5. Dotted circle indicates the region of the network chosen for further investigation. Abbreviations in the chart: SNF-5 (Sodium: Neurotransmitter symporter Family, orthologous to SLC6A8), BTD (biotinidase), MCT-1 (Mono carboxylate Transporter family, orthologous to SLC16A14), BCAT-1 (branched amino acids transporter), DLD (dihydrolipoamide dehydrogenase), FPGS (folylpolyglutamate synthase), TYMS-1 (thymidylate synthetase), MEL-32 (orthologue of SHMT1 serine hydroxymethyl transferase 1) THK-1 (thymidine kinase-1). **d** Quantitation of folic acid intermediates using targeted metabolomic analysis in longevity mutants (day 1 adult). FA, THF, and ML accumulate in three out of four longevity mutants. 5MTHF significantly decreases in three longevity mutants. Abbreviations in the chart: FA (folic acid), DHF (dihydrofolic acid), THF (tetrahydrofolic acid), 5MTHF (5-methyl-tetrahydrofolic acid), ML (5,10-methylene-tetrahydrofolic acid), MN (5,10-methenyl-tetrahydrofolic acid), FO (formyl-tetrahydrofolic acid). **a**, **d** $N = 5$ independent biological replicates. **a**, **d** Normalized metabolite concentrations are converted to log2 for heat map generation. **a**, **b** Statistics were performed using one-sided Fisher test and Benjamini–Hochberg correction for multiple comparisons (adj *p* < 0.05) and **d** using one-way-ANOVA and Dunnett's multiple comparison *p* < 0.5, **p* < 0.01, ***p* < 0.001 (Supplementary Table 6, for statistics).

health and the large degree of differential regulation we observed in our data (Fig. 1a–c and Supplementary Data 1), we decided to follow up on the possible role of these metabolites in longevity.

**One carbon metabolism folate cycle is altered in longevity mutants.** One-carbon metabolism mediated by folate cofactors supports multiple physiological processes including amino acid homeostasis (methionine, glycine and serine), biosynthesis of nucleotides (purines, thymidine), epigenetic maintenance, and redox defense[13–15]. Folate is obtained from the diet and is converted to tetrahydrofolate (THF), which serves as the backbone for one-carbon reactions. Enzymes of the folate cycle catalyze the various reactions that transition carbon through three different oxidation states typified by 5,10-methenyl-tetrahydrofolic acid (MN), 5,10-methylene-tetrahydrofolic acid (ML), and 10-formyl-tetrahydrofolic acid (FO)[16,17] (Fig. 1d). Many such oxidation and reduction steps are NADPH/NADP dependent, and the folate cycle is actually a major generator of cellular NADPH.

Dihydrofolate reductase (DHFR-1) carries out the first two reaction steps, reducing folic acid to dihydrofolic acid (DHF) and on to tetrahydrofolate (THF) (Fig. 1d). Using serine as a methyl donor, serine hydroxymethyl transferase 1 (MEL-32) then produces 5,10-methylene-tetrahydrofolic acid (ML), a key hub intermediate. Thereafter methylene tetrahydrofolate reductase (MTHFR-1) converts ML to 5MTHF, feeding into the methionine cycle (and reconstituting THF in the process). Methylene tetrahydrofolate dehydrogenase (DAO-3) converts ML to MN, and on to 10-formyl-tetrahydrofolic acid (FO)[16]. Phosphoribosylglycinamide formyl-transferase (GART) ortholog F38B6.4 transfers the formyl group from FO for use in purine biosynthesis (and converts FO back to THF in the process). Thymidylate synthase TYMS-1 also uses ML to convert dUMP to dTMP for DNA synthesis (and restores DHF in the process). DHFR also acts closely with TYMS-1 to recycle DHF back to THF. Notably, DHFR1 and TYMS1 are intimately linked as bifunctional enzymes in parasites, and copurify in plants, and together represent the rate-limiting enzymes for one-carbon metabolism[18–20].

Because our untargeted metabolomic analysis revealed a significant decrease of 5 methyl tetrahydrofolate (5MTHF) in three out of four longevity mutants, we subsequently performed targeted metabolomics and quantified all major folic acid forms, except those conjugated to glutamate. We observed that FA, THF, and ML were more abundant in *daf-2*, *eat-2*, and *isp-1* by 2 to 4-fold. 5MTHF and MN were 2-fold lower in these three genetic backgrounds, while FO was decreased in 3 of 4 genotypes (Fig. 1d and Supplementary Table 6). Additionally, THF was significantly

elevated in *glp-1* mutants. These results show that metabolites of the FA pathway undergo extensive quantitative changes in multiple long-lived strains, and that 5MTHF, in particular, is robustly and reproducibly reduced.

**Folate cycle genes *dhfr-1* and *tyms-1* regulate life span.** As we observed a common accumulation of folic acid in several long-lived worm mutants, we next asked if dietary supplementation of this compound influenced life span. Labeled FA entered the worm (Supplementary Fig. 3a), yet feeding physiological concentrations of FA (10 nM) had no effect on WT life span, pharyngeal pumping rate, or brood size (Supplementary Fig. 3b–d). Bacteria are the main dietary source of folates for worms, but supplementation of submolar FA concentrations did not significantly affect levels of bacterial folate pathway intermediates (Supplementary Fig. 3e, f).

To manipulate the intracellular concentration of folic acid intermediates in vivo, we utilized RNAi against key enzymes of the pathway combined with supplementation experiments. We focused on folate cycle enzymes and first performed a mini-screen using RNAi against *dhfr-1*, *mel-32*, *mthf-1*, *dao-3*, *tyms-1*, and *F38B6.4* from L4 stage on (Supplementary Fig. 3g). RNAi knockdown of these genes reduced mRNA expression to 20–60% of controls (Supplementary Fig. 3h). Only *tyms-1i* and *dhfr-1i*, however, increased mean life span significantly, by 25% and 28%, respectively (Supplementary Table 7). Further ageing experiments confirmed that *dhfr-1* RNAi administered from L4 stage on consistently extended worm mean life span by 17–26% (Fig. 2a and Supplementary Table 7), with little effect on food intake as measure by pharyngeal pumping rates or reproductive capacity as measured by brood size (Supplementary Fig. 3i, j). Similarly, RNAi knockdown of *tyms-1* also led to a 20% increase in worm median life span (Fig. 2a and Supplementary Table 7), consistent with its close association with *dhfr-1*.

We then measured the levels of folate cycle intermediates upon *dhfr-1i* knockdown by targeted metabolomics (Fig. 2c and Supplementary Table 6). *dhfr-1i* led to an accumulation of upstream FA, concomitant with a reduction in downstream intermediates THF, 5MTHF, MN, ML, and FO, confirming our previous results[21]. Consistent with a critical role as a rate-limiting enzyme in the one-carbon cycle, supplementation of *dhfr-1i*-treated worms with the downstream intermediate 5MTHF restored near-normal levels of all folate cycle intermediates, except for upstream FA, which remained high. In *luci* controls, dietary 5MTHF also elevated levels of THF, 5MTHF, MN beyond untreated animals, but had no effect on levels of ML or FO

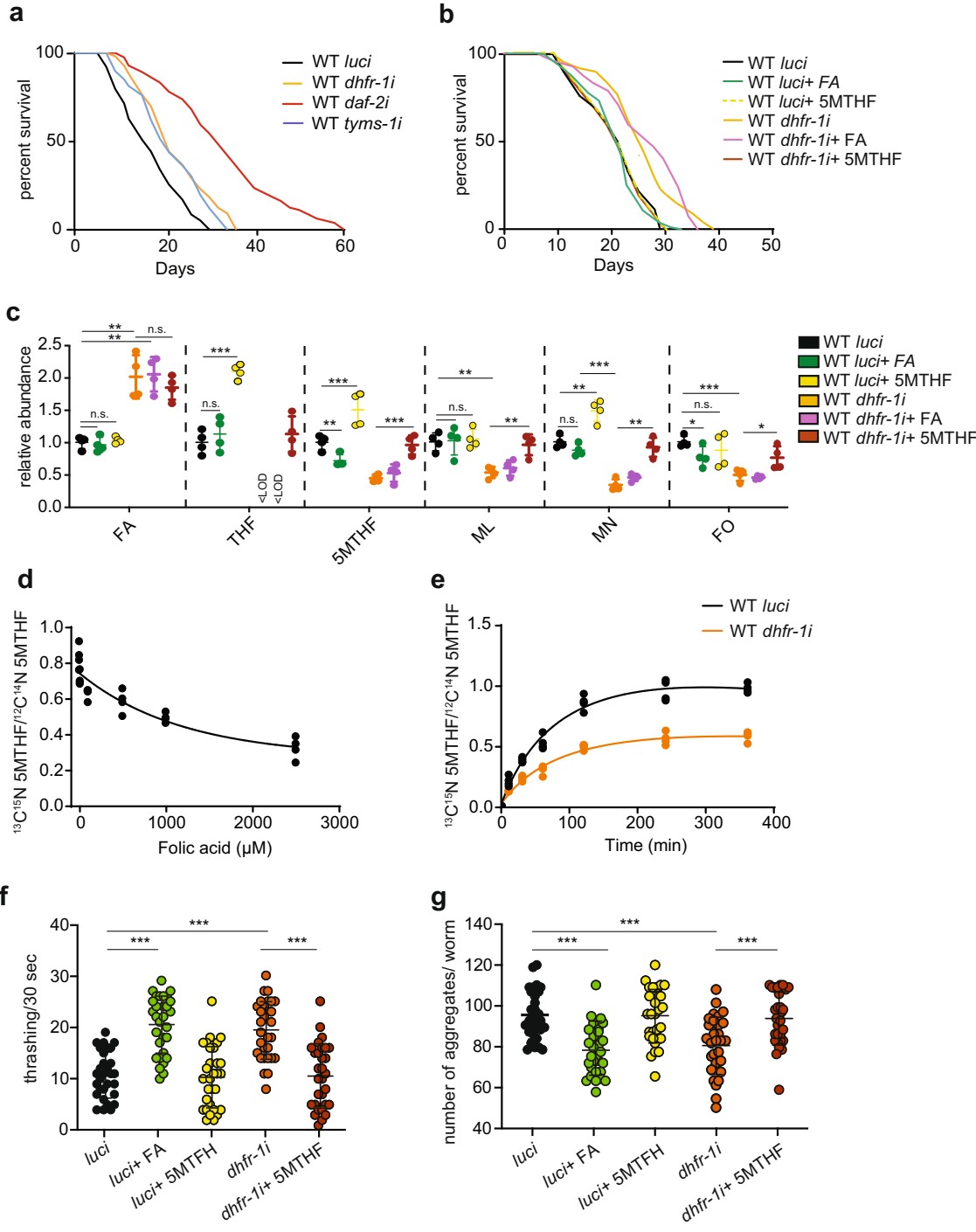

**Fig. 2 *dhfr-1i* prolongs nematode life span in a 5MTHF dependent manner. a** *dhfr-1* and *tyms-1* RNAi treatment increase wild-type life span (from L4 stage). **b** Supplementation of 10 nM 5MTHF abolishes *dhfr-1i* longevity (from L4 stage). **c** Quantitation of folic acid intermediates using targeted mass spectrometry in wild-type worms (day 1) with *luci* and *dhfr-1i* treatment in the presence or absence of folic acid and 5MTHF. *dhfr-1i* increases folic acid and decreases 5MTHF and downstream intermediates. **d** Increasing concentrations of FA inhibit ${}^{13}C{}^{15}N$ labeled 5MTHF incorporation within a 2 h time period (day 1 adult). **e** Incorporation of ${}^{13}C{}^{15}N$ labeled 5MTHF over time in worms with *luci* (black line) and *dhfr-1i* (orange line) (day 1 adult) each dot represents a single biological replicate. **f** Thrashing assay of polyQ35 worms (day 7 adult). **g** Protein aggregate quantitation in polyQ40 model (day 7 adult). *dhfr-1i* or FA supplementation are beneficial, whereas 5MTHF is detrimental for motility and aggregate accumulation. Each dot represents a single worm. **a**, **b** $n = 150$ worms per repeat per condition, $N = 3$ biological replicates. **c** $N = 5$ biological replicates. **d**, **e** $N = 4$ independent biological replicates. **f**, **g** $n = 30$ worms, $N = 3$ independent biological replicates, only one biological replicate is shown. **a**, **b** Statistics were performed with the two-sided Mantel–Cox log-rank test (Supplementary Table 7 for statistics). **c**, **f**, **g** Significance was assessed using one-way ANOVA and Dunnett's multiple comparisons test. $*p < 0.5$, $**p < 0.01$, $***p < 0.001$. Data are presented as mean ± S.D. (Supplementary Table 8 for statistics).

(Fig. 2c and Supplementary Table 6). On the other hand, supplementation of folic acid to *dhfr-1i* did not alter any folate intermediates. Given that 5MTHF uptake by the worm largely restored folate pools, we next asked whether supplementation would impact longevity. Whereas 10 nM 5MTHF supplementation had little effect on *luci* controls, it abolished extension of life by *dhfr-1i* (Fig. 2b and Supplementary Table 7), indicating that *dhfr-1i* longevity arises from lower levels of 5MTHF (or altered levels of other folate intermediates).

**FA regulates 5MTHF uptake**. Interestingly, we noticed that *dhfr-1i* led to the accumulation of FA, and that FA supplementation led to similar changes in folate intermediates as *dhfr-1i* knockdown in the worm, though not to the same extent. Specifically, both conditions led to a lowering of 5MTHF, MN, and FO (Fig. 2c). This suggested that the FA-induced decrease in these downstream intermediates might arise via inhibition of uptake or negative feedback. To test this idea, we measured the incorporation of 5MTHF in adult worms, by quantifying the uptake ratio of $^{13}C$ $^{15}N$ double-labeled versus non-labeled 5MTHF. We found that increasing concentrations of FA diminished labeled 5MTHF incorporation in vivo (Fig. 2d). Similarly, *dhfr-1* RNAi reduced the rate of 5MTHF uptake and assimilation over time (Fig. 2e). These findings suggest an inhibitory role of FA in the assimilation of folate cycle intermediates, supporting previous in vitro work[22,23].

**High FA and low 5MTHF ameliorate models of polyQ proteotoxicity**. Many long-lived strains exhibit improved protein homeostasis that is often manifested as greater resistance to toxic aggregate-prone proteins during aging. Various polyglutamine repeats expressed in *C. elegans* muscle, modeling Huntington's disease, form aggregates and induce age-related progressive paralysis, with severity and age of onset related to repeat length[24]. To explore other possible benefits of FA supplementation or *dhfr-1i* treatment, we investigated the effect of folates on polyQ repeat proteotoxicity models. Whereas supplementation of FA or endogenous accumulation of FA via *dhfr-1i* significantly enhanced the motility of polyQ35 worms, supplementation with 5MTHF in both conditions was detrimental (Fig. 2f). We also tested the polyQ40 proteotoxicity model, and found that *dhfr-1i* reduced the number of visible protein aggregates, while 5MTHF supplementation brought such aggregates back to WT levels (Fig. 2g). These findings suggest that elevated FA and lower 5MTHF ameliorate proteotoxicity and improve healthspan.

***dhfr-1i* induces methionine restriction**. 5MTHF provides essential substrates for methionine synthase (MS) and is the main source of carbon units for the methionine cycle[13]. Homocysteine captures the methyl group from 5MTHF to form methionine and is rapidly converted to *S*-adenosyl methionine (SAM). SAM then donates its methyl group to various molecular acceptors, and in the process generates adenosyl-homocysteine, which is then converted to homocysteine. Methylation of homocysteine completes the methionine cycle (Fig. 3a). Aside from the methionine cycle, homocysteine can be channeled into transsulfuration, glutathione, and pyruvate pathways.

We next investigated how levels of methionine cycle intermediates varied upon *dhfr-1i* and 5MTHF feeding using targeted metabolomics. *dhfr-1i* resulted in significantly lower levels of methionine and *S*-adenosyl methionine. *dhfr-1i* or FA supplementation also resulted in the accumulation of homocysteine and adenosyl-homocysteine (Fig. 3b, c and Supplementary Table 6). Supplementation of 10 nM 5MTHF was sufficient to significantly

restore methionine levels in *dhfr-1i*, and reduce levels of homocysteine and *S*-adenosyl-homocysteine.

*metr-1* encodes the *C. elegans* homolog of methionine synthase, and *metr-1i* induces methionine restriction in the worm[16,25]. A comparison of methionine cycle intermediates in *dhfr-1i* and *metr-1i* revealed several similarities, including reduced methionine, *S*-adenosyl methionine, and increased homocysteine (Supplementary Fig 4a, b and Supplementary Data 2). Thus, *dhfr-1i* not only reduced 5MTHF, but also perturbed the methionine pool in a manner similar to *metr-1i*. *dhfr-1i* and *metr-1i* also shared a more extensive metabolomic signature. Twenty-two metabolites were found in common ($p < 0.05$), including those known to change upon methionine restriction, such as tryptophan, methionine sulfoxide, nucleosides, uracil, and thymidine as well as metabolites associated with energy metabolism[26,27] (Supplementary Fig. 4c, d and Supplementary Data 2).

In mammalian cells, methionine restriction induces a characteristic transcriptional profile that can be used to monitor the process[28]. We first used *metr-1i* to characterize gene expression changes upon methionine restriction in the worm and validated that several nematode orthologues of genes found in the mammalian study were similarly regulated (Supplementary Table 8). We then characterized *dhfr-1i* knockdown, and found that it induced a similar signature, causing a significant 1.5-2-fold increase in the expression of these genes similar to *metr-1i* (Fig. 3d). This increase was specific, as 5MTHF supplementation of *dhfr-1i*-treated animals reverted expression to control levels (Fig. 3e) thereby indicating a specific gene response resembling methionine restriction[27,28].

To test the idea that methionine limitation is causal for *dhfr-1i* longevity, we next performed methionine supplementation experiments. Notably, we observed a dose-dependent reduction in *dhfr-1i* longevity upon 20 and 40 mM methionine supplementation, with 40 mM fully suppressing long life (Fig. 3f and Supplementary Table 7). Similar supplementation of *luci* controls had little effect. These findings confirm that *dhfr-1i* induces longevity via methionine restriction.

To investigate more globally how *dhfr-1i* affects other metabolic pathways, we performed untargeted metabolomics on *dhfr-1i* with and without 5MTHF supplementation (Supplementary Fig. 4e, f and Supplementary Data 2). In particular, we found that changes in methionine sulfoxide, uracil, thymidine as well as components of spermine and energy metabolism were partially or fully reversed upon 5MTHF supplementation ($p < 0.05$).

These data reveal possible novel connections between the one-carbon metabolism and other metabolic processes.

**5MTHF affects the longevity of long-lived IIS and mitochondrial mutants**. Our original metabolomic analysis indicated that *C. elegans* longevity mutants, *daf-2* and *isp-1* exhibit a prominent reduction in 5MTHF and other FA intermediates (Fig. 1d). We asked whether lower levels of 5MTHF play a role in life span regulation in these mutants by supplementing 5MTHF. 5MTHF treatment modestly reduced mean and max life span of *daf-2* (mean: −15.8%, max: −22%) and *isp-1* (mean: −13%, max: −15%), suggesting that lower levels of the 5MTHF contribute partially towards longevity in these mutants (Fig. 4a, b and Supplementary Table 7).

To further investigate the cause of the lower 5MTHF level in these two genotypes, we quantified the mRNA expression of various folic acid cycle genes. Interestingly, *dhfr-1* mRNA expression was decreased by more than 50% in both mutants. *mtfh-1* expression was also ca. 50% lower in the *daf-2* background (Fig. 4c, d). Other folic acid cycle genes (*mthr-1*, *mel-32*, *dao-3*,

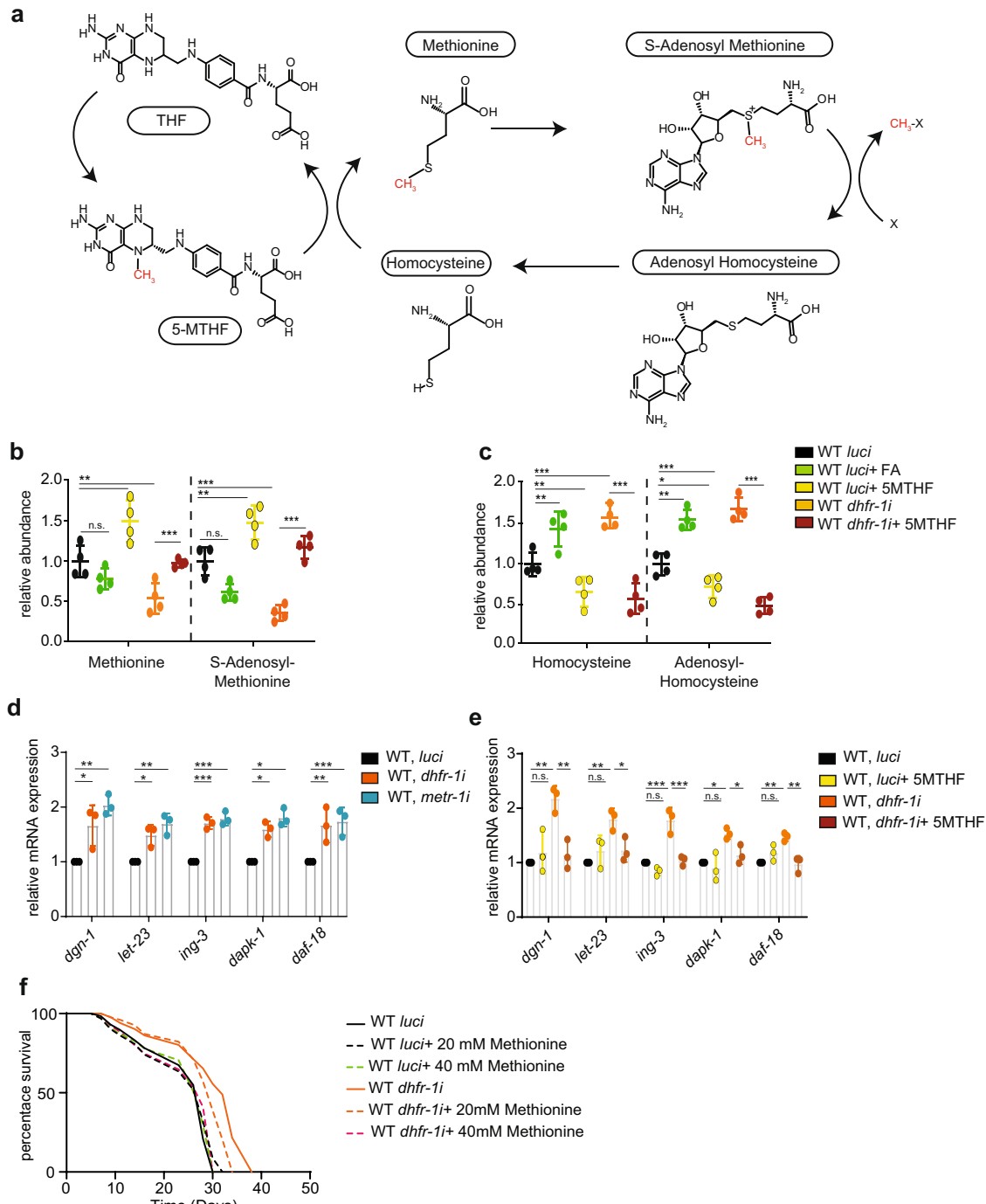

**Fig. 3 *dhfr-1i* affects the methionine cycle and mimics methionine restriction. a** Schematic of the methionine cycle. **b**, **c** Quantitation of methionine cycle intermediates using targeted mass spectrometry in *dhfr-1i* and *luci*, with or without FA and 5MTHF supplementation (day 1 adult). *dhfr-1i* decreases methionine and *S*-adenosyl methionine levels and increases homocysteine and adenosyl-homocysteine levels. This increase is reversed by supplementation with 5MTHF. **d** *dhfr-1i* and *metr-1i*. show similar regulation of mRNA expression of selected genes implicated in methionine restriction (*dgn-1, let-23, ign-3, dapk-1,* and *daf-18*), day 1 adults. **e** Methionine restriction specific gene signature is increased by *dhfr-1i* and reversed by 5MTHF supplementation, day 1 adults. **f** Methionine supplementation (20 mM, 40 mM) suppresses *dhfr-1i* longevity. **b**, **c** $N = 5$ independent biological replicates. **d**, **e** $N = 3$ independent biological replicates. **f** $n = 150$ worms per repeat per condition, $N = 3$ biological replicates. **b–e** Significance was assessed using one-way anova with Dunnett's multiple comparisons test *$p < 0.5$, **$p < 0.01$, ***$p < 0.001$. **f** Statistics were performed with the two-sided Mantel–Cox log-rank test. *$p < 0.5$, **$p < 0.01$, ***$p < 0.001$. Data are presented as mean ± S.D. (Supplementary Table 8 for statistics).

and *F38B6.4*) were unchanged. Further, we found that *dhfr-1* mRNA expression was also downregulated in the dietary restriction model *eat-2* relative to wild-type based on published RNA-seq data[29] (Supplementary Fig. 5a).

The DAF-16/FOXO winged helix transcription factor is a major regulator of longevity whose mutation negates life extension in both *daf-2* and *isp-1* pathways[30,31]. In response to reduced insulin/IGF or mitochondrial signaling, DAF-16 localizes

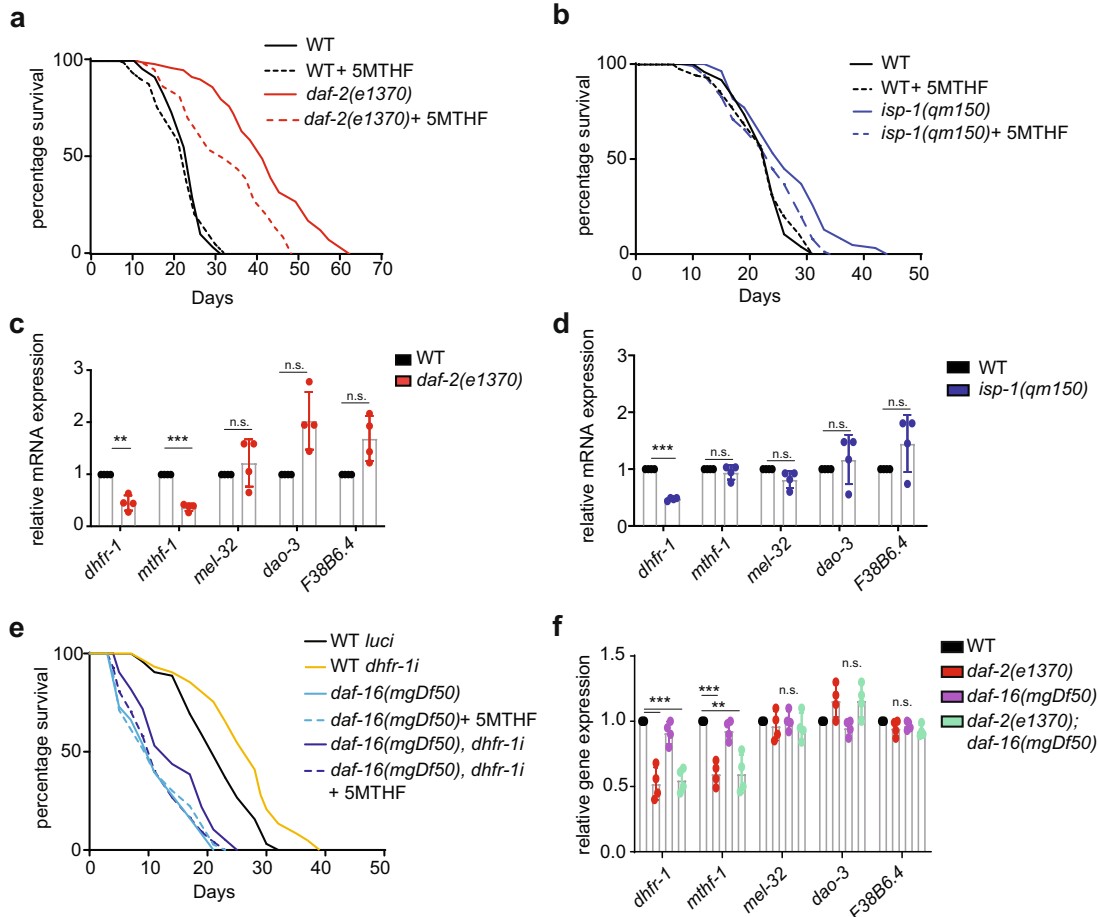

**Fig. 4 dhfr-1 and 5MTHF act within insulin/IGF and mitochondrial signaling pathways. a** Life span of daf-2(e1370) supplemented with 10 nM 5MTHF (from L4 stage). **b** Life span of isp-1(qm150) supplemented with 10 nM 5MTHF (from L4 stage). 5MTHF does not alter WT life span but reduces daf-2 and isp-1 longevity. **c**, **d** Relative mRNA expression of folate cycle genes in daf-2(e1370) and isp-1(qm150) (day 1 adult). dhfr-1 mRNA expression is lower in both genotypes. **e** Life span experiment of wild-type and daf-16(mgDf50) with luci and dhfr-1i in the presence or absence of 5MTHF (from L4 stage). dhfr-1i extends daf-16 mutant life span and supplementation with 5MTHF abolishes this extension. **f** Relative mRNA expression of folate cycle genes in daf-16 and daf-16,daf-2 backgrounds. daf-16 mutation has little or no effect on dhfr-1 or mthf-1 mRNA expression in the daf-2 background. **a**, **b**, **e** $n = 150$ per repeat per condition, $N = 3$ independent biological replicates. **c**, **d**, **f** $N = 3$ biological replicates. **a**, **b**, **e** Statistics were analyzed by the two-sided Mantel–Cox log-rank test (Supplementary Table 7 for statistics). **c**, **d**, **f** Significance was assessed using one-way ANOVA and Dunnett's multiple comparisons test. *$p < 0.5$, **$p < 0.01$, ***$p < 0.001$. Data are presented as mean ± S.D. (Supplementary Table 6 for statistics).

to the nucleus to control transcription of target genes involved in oxidative stress response, heat shock, and lipogenesis[32]. Because we found that dhfr-1 mRNA levels were regulated by daf-2 and isp-1 (Fig. 4c, d), we asked if dhfr-1i life span extension also showed daf-16 dependence. Aging experiments showed that the median life span of daf-16(mgDf50) was modestly increased upon dhfr-1i by 6–8% (Fig. 4e and Supplementary Table 7). This increase was reversed upon supplementation with 5MTHF. We next asked whether dhfr-1i affects daf-2 longevity. Aging experiments showed that dhfr-1i did not further extend the mean and maximum life span in the daf-2 background (Supplementary Fig. 5c and Supplementary Table 7). These findings suggest that dhfr-1i life span extension might act within the insulin/IGF pathway, downstream or parallel to daf-16. To further address this idea, we examined dhfr-1 mRNA expression in daf-2, daf-16, and daf-16;daf-2 double mutants. We saw that daf-16 mutation either on its own or in the daf-2 background had little or no effect on dhfr-1 and mthf-1 mRNA expression (Fig. 4f). We also examined whether dhfr-1 impacts daf-16 dependent genes, gst-4, dod-3 and sod-3[33]; but their expression was unchanged upon dhfr-1i and 5MTHF treatment (Supplementary Fig. 5b). Altogether, these findings indicate that dhfr-1 is downregulated in

response to reduced insulin/IGF signaling, but in a manner independent of daf-16. Thus, the regulation of dhfr-1 likely works through other transcription factors.

**Conserved regulation of folate and methionine cycle intermediates.** Because dhfr-1 affected the methionine cycle (Fig. 3), and dhfr-1 and daf-2 had overlapping longevity phenotypes (Supplementary Fig. 5c) we wondered whether daf-2 mutation also perturbs methionine metabolism. To address this, we performed targeted metabolomics. We saw that, like dhfr-1i, daf-2 mutation resulted in significantly higher levels of homocysteine and adenosyl-homocysteine, lower levels of S-adenosyl methionine, as well as a trend towards lower levels of methionine, though the latter did not reach significance (Fig. 5a and Supplementary Table 6). Furthermore, a subset of genes induced by methionine restriction was also induced in daf-2 mutants (Fig. 5b). None of these gene expression changes, however, showed daf-16 dependence.

The insulin receptor substrate protein 1 (Irs1) is a key mediator of insulin/IGF signaling (IIS) in mammals. Homozygous disruption of Irs1 in mice results in life span extension, resistance to

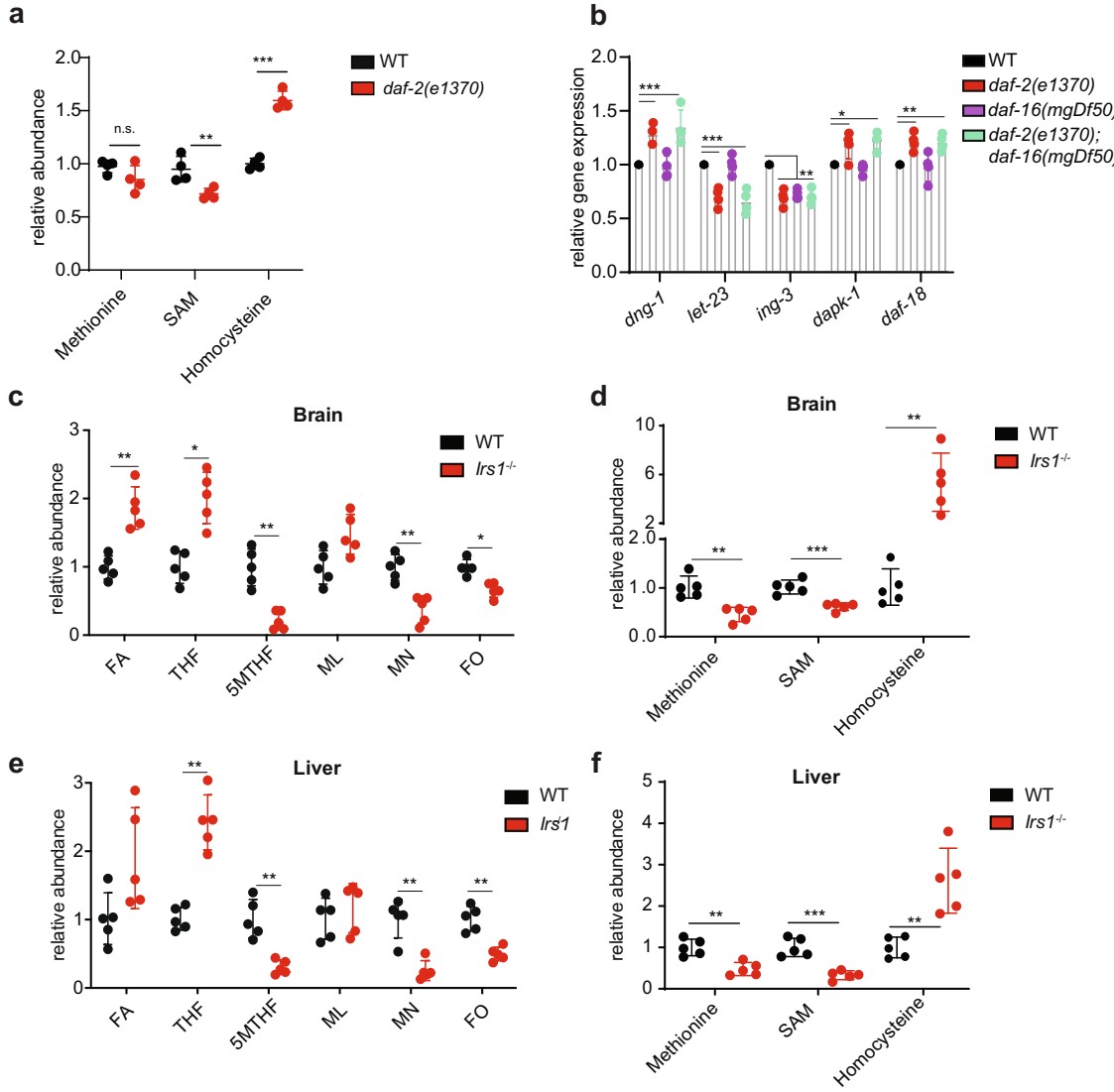

**Fig. 5 Reduced insulin/IGF signaling alters folic acid and methionine cycle intermediates. a** Relative abundance of methionine cycle intermediates in *daf-2* mutants and WT animals using targeted metabolomics. **b** Relative mRNA expression level of methionine restriction gene signature in *daf-2* and *daf-2; daf-16* backgrounds. *dgn-1, ing-3, dapk-1,* and *daf-18* are upregulated and *let-23* and *ing-3* downregulated in the *daf-2* background, independent of *daf-16*. **c, e** Targeted metabolomics of folic acid intermediates in brain and liver in wild-type and $Irs1^{-/-}$ full body knockout mice. Folic acid and THF increase in $Irs1^{-/-}$, whereas 5MTHF decreases in the mutant. **d, f** Quantitation of methionine cycle intermediates in brain and liver in wild-type and $Irs1^{-/-}$ knockout mice. Methionine and SAM decrease and homocysteine increases in the mutant. **a** $N = 4$ independent biological replicates. **b** $N = 3$ independent biological replicates, **c–f** $N = 5$ mice, each dot represents a single animal. **a–f** Significance was assessed using one-way ANOVA Dunnett's multiple comparisons test. *$p < 0.5$, **$p < 0.01$, ***$p < 0.001$. Data are presented as mean ± S.D. (Supplementary Table 6 for statistics).

several age-related pathologies including bone and motor dysfunction, skin, and protection against glucose intolerance[34,35]. Because reduced insulin/IGF signaling in worms altered folate and methionine homeostasis (Figs. 1d and 5a), we wondered if similar changes occur in the $Irs1^{-/-}$ mouse model. Targeted metabolomic analysis of folic acid intermediates using brain and liver tissues from female $Irs1^{-/-}$ knockout mice revealed a profile similar to worm *daf-2* mutants (Fig. 5c, e and Supplementary Table 6). THF was significantly increased up to 2-fold in both brain and liver in $Irs1^{-/-}$, while FA, and ML also tended to increase though these did not always reach significance. In addition, 5MTHF, MN, and FO were significantly decreased in both tissues. We also quantified methionine cycle intermediates and observed significantly lower levels of methionine and *S*-adenosyl methionine, and a large increase in homocysteine in these tissues (Fig. 5d, f and Supplementary Table 6). Altogether

these data suggest that the insulin/IGF signaling pathway similarly regulates the folate and methionine cycle from worms to mammals.

## Discussion
Folates are essential vitamins derived from dietary sources and the microbiome. The folate cycle provides one-carbon units for an extensive metabolic network that fuels the methionine cycle, transsulfuration pathway, de novo purine synthesis, thymidine production, serine, glycine, glutathione, and NADPH pools, and thereby regulates cellular redox state, growth, and proliferation[16,36]. The major canonical longevity pathways also profoundly regulate metabolism, growth, and organismal physiology in numerous ways[37]. By generating the metabolic profiles of several different long-lived mutants in *C. elegans*, we discovered that folate cycle

intermediates represent a convergent focal point for longevity regulation across conserved signaling pathways.

Generally, we found that several folate intermediates were consistently altered. 5 methyl tetrahydrofolate levels were specifically decreased in long-lived models of downregulated insulin/IGF signaling (*daf-2*), mitochondrial respiration (*isp-1*), and dietary restriction (*eat-2*), while THF was upregulated in reproductive signaling (*glp-1*) mutants. In accord with a physiological role in longevity, knockdown of folate cycle enzyme *dhfr-1* in the wild-type background was sufficient to similarly decrease 5MTHF, extend life span and reduce polyQ proteotoxicity. Supplementation with this metabolite restored the folate pool and reversed these phenotypes. Consistently, *daf-2*/InsR and *isp-1*/Rieske longevity were reduced by 5MTHF supplementation, and these mutant strains downregulated levels of *dhfr-1* mRNA, supporting regulation of folate metabolism by these signaling pathways. Thus, reduced 5MTHF and other intermediates of folate are causal and/or associated with longevity as part of a shared mechanism in several pathways.

Previous studies in *C. elegans* have described various effects of FA on longevity. The anti-diabetic drug metformin extends *C. elegans* life span by disrupting microbial folate production and inducing methionine restriction[38], while inhibition of endogenous *C. elegans* genes for folate uptake or folate polyglutamase activity had little effect on longevity[39]. By contrast, Rathor and colleagues reported that μM levels of FA extend life span[40]. We saw no effect of 10 nM FA supplementation on worm life span, but observed an improvement in polyglutamine proteotoxicity models, suggesting folates as potential therapeutics for proteoprotection[41]. These seemingly disparate observations probably reflect differences in compound dose and availability on uptake or feedback, as well as the complex interplay between diet, microbiome, and host[16,25]. In our case, we targeted rate-limiting interlinked enzymes of the *C. elegans* folate pathway, *dhfr-1* and *tyms-1*, and saw coherent changes in folate intermediates and extension of worm life span. Supplementation with nanomolar amounts of 5MTHF were sufficient to restore folate pools of *dhfr-1i*, with little effect on bacterial folate pools. Knockdown of other folate cycle enzymes might not have elicited these phenotypes in our hands because of RNAi efficiency or pleiotropic effects on other processes. Interestingly, a link between longevity and folate metabolism has been recently reported in budding yeast: Rpl22 ribosomal protein mutants increase yeast replicative life span, show lower levels of metabolites associated with folate, serine, and methionine metabolism, and deletion of 1C enzymes enhanced wild-type longevity[42]. Interestingly, folate supplementation has also been shown to modulate the DNA methylation aging clock[14].

What is the mechanism by which *dhfr-1i* and 5MTHF reduction trigger longevity? Several lines of evidence argue that it acts through methionine restriction (MR) and modulation of methionine cycle intermediates. First, we observed that *dhrf-1* knockdown perturbs the methionine cycle, leading to reduced levels of methionine and *S*-adenosyl methionine and elevated levels of homocysteine. Second, we observed changes in gene expression associated with this process. Third, a comparison of metabolomic profiles from *dhfr-1i* and *metr-1i* methionine synthase revealed significant overlap. Fourth, methionine supplementation of *dhfr-1i* reversed longevity in a dose-dependent manner. Altogether these findings indicate that methionine restriction is causally linked to *dhfr-1i* life extension. Strikingly we saw similar changes in folate and methionine pools in tissues of *Irs1*−/− knockout mice and *daf-2*/InsR mutant worms, revealing that the control of the folate cycle by insulin/IGF signaling is evolutionarily conserved. Whether these changes are causally connected to life span regulation in mammals remains to be seen.

Though the metabolic link between 5MTHF and methionine seems clear, it should be pointed out that 10 nM 5MTHF and 40 mM methionine were required for *dhfr-1i* rescue, showing large differences in concentration. These concentration levels are consistent with the literature[43,44], and probably reflect differences in the uptake or utilization of these compounds, but also raise the possibility that other 5MTHF/methionine derived metabolites or substrates could be involved. It also seems likely that additional amino acids other than methionine regulate the longevity of *dhfr-1i*, namely glycine and serine, which act in the folate cycle[16]. In this analysis, however, we were not able to measure levels of these amino acids, and thus cannot exclude their contribution in our proposed model.

Methionine restriction has a profound effect on physiology and regulates longevity through multiple mechanisms. First described to prolong life span in rodents, it shows overlapping but also distinct features from other longevity models[45,46]. Notably, MR reduces adiposity, IGF-1, insulin, thyroid hormone levels, increases stress resistance, energy expenditure, insulin sensitivity as well as adiponectin and FGF21 levels[47]. In concert with these changes, MR reportedly alters mitochondrial function, increasing aerobic capacity, and fatty acid oxidation utilization, as well as affecting redox state, reducing mitochondrial ROS production and altering glutathione and peroxidase levels[48]. Further, it can prolong life span in progeria models, and delay age-related decline in immune and cardiovascular function, as well as reduce cellular senescence in the kidney[45]. More recent work has suggested possible benefits of MR on human health including obesity, cancer, and various serum biomarkers[49–51].

MR also promotes longevity in invertebrate models. In *Drosophila*, MR stimulates longevity under low but not high amino acid status, possibly via mTOR signaling[52]. Flies with reduced IIS show evidence of methionine restriction and dependence on enzymes involved in the methionine cycle for longevity[53]. In *C. elegans*, the biguanide anti-diabetic drug, metformin, reduces microbial folates in their bacterial food, resulting in methionine restriction and life span extension dependent on *aak-2*/AMPK and *skn-1*/NFE-2[25,54]. While *metr-1* mutation itself had little effect on life span, it enhanced the life-extending properties of metformin. In yeast, MR has been shown to extend chronological life span, regulate the mitochondrial retrograde response, amino acid general control pathways, and mTOR signaling as well as enhance autophagy and mitophagy[55–57].

SAM metabolism ramifies into multiple intermediates and pathways associated with longevity. Strikingly, we observed reduced levels of SAM in all four longevity pathways including *glp-1*, as well as *Irs1*−/− knockout mice, suggesting a convergent mechanism (though levels of SAM are likely controlled through different routes in *glp-1* given the differences in 5MTHF regulation). As a potent methyl donor, SAM catalyzes the methylation of rRNA, DNA, epigenetic factors, and aids spermidine synthesis, and thus could impact longevity at multiple levels[55,58]. Indeed, mutants in *sams-1* and *sams-5* have been shown to extend worm life span, while tissue-specific downregulation of similar enzymes in *Drosophila* enhances health and life[52,59,60]. Another methionine cycle intermediate, homocysteine, can be diverted to transsulfuration reactions, producing cysteine, $H_2S$, and glutathione, and itself can modulate insulin/IGF signaling[61,62]. In multiple species, sulfur amino acid limitation activates transsulfuration pathways, leading to changes in the production of $H_2S$, a signaling gas that stimulates survival mechanisms[63]. Clearly then folate and methionine cycle metabolism have broad effects on the physiology of ageing[36,47,64–68], and manipulation of pathway enzymes and metabolites may provide new entry points to enhance health and longevity.

Metabolomics has emerged as a powerful approach to identify not only markers of health, disease, and ageing[68], but also causal mechanisms[69]. Our profiles of the different longevity pathways can now be cross-referenced with other studies to generate testable hypotheses, and serve as a useful resource for the field. Interestingly, our analysis revealed congruent changes in other crucial metabolic pathways that could additionally contribute to life extension[10]. Among the changes, we saw modulation of kynurenine and nicotinamide metabolism, which are biochemically interlinked, and have been previously shown to regulate longevity[70,71]. We detected consistent changes in various nucleoside-related metabolites involved in the nucleic acid synthesis and signaling, some of which have been also associated with longevity (thymine, adenosine[72,73]). As well, we observed changes in propionyl carnitine, which impacts acyl-CoA and succinate metabolism, and gamma-glutamyl cysteine involved in glutathione metabolism[74]. We also saw changes in 2-amino benzoic acid in three longevity mutants. Derived from tryptophan by the action of the kynurenine pathway, this metabolite has been suggested as an endogenous marker of organismal mortality in the nematode[75]. We also observed many interesting changes in the metabolome upon *dhfr-1i*. Some changes were expected (e.g., folate intermediates, methionine cycle, pyrimidine, polyamine metabolism, carnitines)[76]. Other changes were unexpected (e.g., AMP, cAMP, NMN, alpha-ketoglutarate, dihydroxyacetone phosphate), suggestive of alterations in TCA, glycolytic, fat, and energy metabolism. Conceivably, some of these metabolic modules could further regulate longevity of reduced insulin/IGF signaling and mitochondrial function, since *daf-2* or *isp-1* mutants had stronger effects on life span than *dhfr-1i*, and 5MTHF supplementation only partially reduced longevity of these two strains. It will be interesting to validate these various changes by targeted metabolomics, and determine the potential roles of these and related metabolites in impacting health and life span.

One limitation of this work is that it only provides a broad snapshot of the steady-state levels of the various metabolites and does not measure the metabolic flux. Whether levels change because of altered synthesis or removal remains to be seen. Additionally, this study focused mainly on polar metabolites, and does not represent an extensive analysis of lipids. Nevertheless, the approach does reveal the power of leveraging multiple longevity pathways to uncover molecules that can impact the aging process and suggest several new hypotheses to be tested.

Our discovery that the regulation of the folate and methionine cycles are convergent mechanisms underlying multiple longevity pathways, and whose regulation by insulin/IGF signaling is conserved in evolution, could provide new ways to improve health during aging.

## Methods

**Worm strains and culture**. All strains were grown and maintained on NGM agar seeded with *E. coli* (OP50) at 20 °C except for the *glp-1(e2141)ts* strain, which underwent a thermal shift to 25 °C leading to germline loss. Because of differing growth rates, worms were harvested for metabolomic analyses after 60 h for WT and *eat-2*, 72 h for *daf-2*, and 145 h for isp-1 worms to ensure similar biological age. Standard procedures for culturing and maintaining strains were used[77]. NGM was prepared using 3 g/L NaCl (Sigma S3014), 2.5 g/L Bacto peptone (BD 211820), 18 g/L Bacto Agar (BD 214010), 25 mM KPO₄, 0.005 mg/mL Cholesterol (Sigma C8667), 1 mM MgSO₄, and 1 mM CaCl₂.

*E. coli* OP50 Bacteria were grown overnight (18 h) in LB media composed of 10 g/L Bacto tryptone (Sigma 95039), 5 g/L Bacto yeast extract (BD 212720), and 5 g/L NaCl.

The complete strain list is provided in Supplementary Table 9.

**RNAi**. Worm RNAi was conducted as described previously[78,79]. Briefly, WT wild-type worms were fed with HT115 (DE3) bacteria transformed with L4440 vector

that expresses a double-stranded RNA against the targeted gene. HT115 strains were diluted prior to seeding till OD 0.2. Synchronized worms were obtained by performing an egg lay on corresponding RNAi plates containing isopropyl-β-D-thiogalactoside and ampicillin. Luciferase (L4440::Luc, i.e., *luci*) RNAi vector was used as a non-targeting control. C36B1.7 (*dhfr-1*), C06A8.1(*mthf-1*), C05D11.11 (*mel-32*), Y110A7A.4 (*tyms-1*), F38B6.4, K07E3.3 (*dao-3*), Y55D5A.5 (*daf-2*), R03D7.1(*metr-1*), RNAi clones were obtained from the Vidal or Ahringer RNAi libraries. The gene targeted by RNAi is indicated with an "i" after the gene name. All RNAi experiments were blinded.

**Folic acid intermediates and methionine supplementation**. Folic acid and 5MTHF acid were added in an aqueous solution into the NGM agar at the indicated final concentrations of 10 nM. For bacteria, folic acid and 5MTHF were added to the medium and incubated 1 or 24 h prior to extraction. Methionine was added in an aqueous solution to NGM agar at the indicated final concentration of 20 or 40 mM.

**Pharyngeal pumping rate assay**. Pharyngeal pumping was assessed by observing the number of pharyngeal contractions during a 30 s interval using twenty synchronized day 1 adult worms in three biological replicates. Experiments were blinded.

**Brood size**. L4 larvae (*n* = 10 animals per strain) were maintained individually under standard conditions. Synchronized young adult worms were singled to 3 cm plates containing OP50. Worms were transferred to fresh plates every 24 h and progeny number counted during a period of 7 days. A minimum of 10 worms were used for each genotype. Experiments were blinded.

**Life span analysis**. Life span analyses were performed at 20 °C as previously reported[78,79]. Data were plotted to calculate mean, median, and maximum lifespans using Microsoft Excel 16.12 and GraphPad Prism 7 Software. For all RNAi life span assays, 150 worms were fed with RNAi from L4 stage on. To determine significance between the life span curves log-rank Mantel–Cox analysis was used. Experiments were blinded.

**RNA extraction and qPCR**. Synchronized day 1 worms (three to four 10 cm plates, ≈8000 worms) were collected in Trizol (Invitrogen). Total RNA was extracted using RNeasy Mini spin column (QIAGEN). The concentration and purity of the RNA was measured by NanoDrop. cDNA was generated using iScript (Bio-Rad). qRT-PCR was performed with Power SYBR Green (Applied Biosystems) on a ViiA 7 Real-Time PCR System (Applied Biosystems). Four technical replicates were averaged for each sample per primer reaction. *cdc-42*, *act-1*, and *amu-1* were used as internal controls. Primers are listed in Supplementary Table 10.

**Fluorescent microscopy**. For fluorescent images of transgenic *C. elegans*, live animals were immobilized with 5 mM sodium azide and mounted on 2% agarose pads. Images were obtained with an Axio Imager Z1 Zeiss microscope. Puncta of the Q40 strain on day 7 were counted from at least 30 worms in three biological replicates. Experiments were blinded.

**Thrashing assay**. Motility was determined by measuring thrashing in liquid. Individual transgenic animals (Q35) on day 7 of adulthood were picked to a 10 μL drop of M9 on a microscope slide and were given a 30 s adjustment period before counting the thrashing rate. Thrashes (defined as the head crossing the vertical midline of the body) were counted for 30 s. A minimal *n*-number of *n* = 30 in three biological replicates was assayed for each genotype. Experiments were blinded.

**Metabolite extraction from worms**. Worm synchronization was achieved using thirty worms to perform an egg lay for two hours on a single plate. Five 10 cm plates (≈10,000 worms) per genotype were combined to obtain one biological replicate. This procedure was repeated five times. Worms in single tubes were washed three times with buffer solution M9 and 0.1% of butylhydroxytoluene (BHT) was added to prevent auto-oxidation as previously reported on our work[21,80]. Samples were snap-frozen in liquid nitrogen and stored at −80 °C before use. Worm pellets were homogenized using a Qiagen tissue lyser for 30 min at 4 °C. Protein concentration was determined using a BCA kit. A volume of worm lysate, which corresponds to 150 μg of proteins for each sample was subjected to Bligh and Dyer extraction (chloroform: methanol, 2:1) for 1 h at 4 °C. Samples were centrifuged at maximum speed for 5 min at 4 °C and supernatant was transferred into a new tube for drying. Before LC injections samples were reconstituted in 10% aqueous acetonitrile. Samples were analyzed using an untargeted method for total metabolomics and for targeted methods to evaluate the abundance of folic acid analogs and methionine cycle intermediates.

**Folic acid intermediates extraction from mouse tissues**. C3B6F1 IRS1 KO and WT control were kindly provided by the lab of Linda Partridge. C57BL/6 *Irs1*⁻/⁻

KO mice were originally obtained from the lab of Prof. Dominic Withers' lab (Imperial College, London). These mice were then backcrossed into the C3H/HeOuJ background by Marker-Assisted Accelerated Backcrossing (MAX-BAX®) conducted by Charles River. In order to generate homozygous C3B6F1 hybrid $Irs1^{-/-}$ KO mice, C3H/HeOuJ $Irs1^{-/+}$ females were mated with males of the C57BL/6 $Irs1^{-/+}$ KO strain. To generate C3B6F1 wild type control mice, C3H/HeOuJ females were mated with C57BL/6NCrl males (strain codes 626 and 027, respectively, Charles River Laboratories). All mice were bred on-site at the mouse facility of the Max Planck Institute for Biology of Ageing, Cologne. The $Irs1^{-/-}$ KO mice were homozygous and only females were used for the experiments. Mouse experiments were performed according to the guidelines and approval of LANUV [Landesamt für Natur, Umwelt und Verbraucherschutz Nordrhein-Westfalen (State Agency for Nature, Environment and Consumer Protection North Rhine-Westphalia), VSG 84-02.04.2014.A215]. Animals were maintained in groups of 5 females in individually ventilated cages under specific-pathogen-free conditions with constant temperature (21 °C), 50–60% humidity, and a 12-h light–dark cycle. The animals were sacrificed at the age of 24 months and tissues were snap-frozen in liquid nitrogen and kept at −80 °C. Tissues underwent similar extraction as described for the worms. Tissues were homogenized using a Qiagen tissue lyser for 30 min at 4 °C. Protein concentration was determined using a BCA kit and the lysate volume corresponding to 150 µg of protein was subjected to Bligh and Dyer extraction (chloroform: methanol, 2:1) for 1 h at 4 °C. Samples were centrifuged at maximum speed for 5 min at 4 °C and supernatant was transferred into a new tube for drying. Before LC injections samples were reconstituted in 10% aqueous acetonitrile. Samples were analyzed using a targeted method to assess the abundance of folic acid and methionine cycle intermediates.

**Untargeted metabolomics.** Analytes were separated using an UHPLC system (Vanquish, ThermoFisher Scientific, Bremen, Germany) coupled to an HRAM mass spectrometer (Q-Exactive Plus, ThermoFisher Scientific GmbH, Bremen, Germany) using a modified RP-MS method from Wang et al.[81] Briefly, two microliters of the sample extract were injected into a X Select HSS T3 XP column, 100 Å, 2.5 µm, 2.1 mm × 100 mm (Waters), using a binary system A water with 0.1% formic acid, B: acetonitrile with 0.1 formic acid with a flow rate of 0.1 mL/min, with the column temperature kept at 30 °C. Gradient elution was conducted as follows: isocratic step at 0.1% eluent B for 0.3 min, gradient increase up to 2% eluent B in 2 min, then increase up to 30% eluent B in 6 min and to 95% eluent B in 7 min, isocratic step at 95% eluent B for 2 min. Gradient decreases to 0.1% eluent B in 3 min and held at 0.1% eluent B for 5 min. Mass spectra were recorded from 100–800 $m/z$ at a mass resolution of 70,000 at $m/z$ 400 in both positive and negative ion modes using data-dependent acquisition (Top 3, dynamic exclusion list 10 s). Tandem mass spectra were acquired by performing CID (isolation 1.5 a.u., stepped collision energy 20 and 80 NCE). The $m/z$ of leucine enkephaline was used as lock mass. The sample injection order was randomized to minimize the effect of instrumental signal drift. MS data analysis was performed using Xcalibur software 4.1.

**Compound identification and quantitation.** Metabolite search was performed using Compound discoverer 2.0 and $m/z$ Cloud as online databases, considering precursor ions with a deviation > 5 ppm, 0.3 min maximum retention time shift, minimum peak intensity 100,000, intensity tolerance 10, FT fragment mass tolerance 0.0025 Da, group covariance [%] less than 30, $p$-value less than 0.05 and area Max greater or equal to 10000. Metabolites are correctly identified when at least two specific fragments are found in the MS$^2$ spectra. Because of the high mass accuracy < 3 ppm, predicted elemental compositions of the unknown features were submitted to other online databases such as Chemspider (http://www.chemspider.com/), HMDB (http://www.hmdb.org/), KEGG (http://www.genome.jp/kegg/), and METLIN (http://metlin.scripps.edu/).

Quantitation was performed using Trace finder 4.0, using genesis detection algorithm, nearest RT, S/N threshold 8, min peak height (S/N) equal to 3, peak S/N cutoff 2.00, valley rise 2%, valley S/N 1.10.

Relative quantitation was obtained by dividing the area of individual metabolite peaks to spiked internal standards prior to extraction (Leucine enkephaline 100 nM (Sigma L9133), myristic acid 50 nM (Sigma M3128), cysteamine-S-phosphate 100 nM (Sigma C8397)).

**Integration of metabolomics features using the network-based algorithm PIUMet.** the identified $m/z$ values that were significantly changed in all genotype comparisons, were uploaded to PIUMet (http://fraenkel-nsf.csbi.mit.edu/piumet2/). We additional included 60 features, which were unidentified but significant in all genotypes (Supplementary Table 4). We used the following parameters: number of trees 10, edge reliability 2, negative prize depth 0.0005, and number of repeats 50.

The Prize-Collecting Steiner Forest algorithm identifies metabolites and represents them as nodes, the higher the assignment score the bigger the node. The algorithm links these features based on high-confidence protein–protein and protein–metabolites interactions using two databases, HMDBv4.0 and Recon3D. Further details such as node frequency and node edge are reported in Supplementary Table 5. The output was processed using the R (Rstudio v.3.3.2)

package "gplot" (v3.0.1) in order to visualize the cluster of metabolites and to highlight the connection between the predicted proteins and metabolites.

**Targeted analysis of folic acid intermediates.** Identification and relative quantitation of folic acid intermediates were performed on a triple quadrupole mass spectrometer (QqQMS) (TSQ Altis, ThermoFisher Scientific GmbH, Bremen, Germany), as previously published by our group, and the validation of the folate identity confirmed using standards[21]. Data were analyzed using Xcalibur version 4.0. Quantitation was performed using Trace finder 4.1, using the genesis detection algorithm, nearest RT, S/N threshold 8, min peak height (S/N) equal to 3, peak S/N cutoff 2.00, valley rise 2%, valley S/N 1.10. The relative response for each folate species was calculated by dividing the peak area of the analyte to the internal standard peak area (pteridinic acid 100 nM, Sigma P1781) and further normalized to protein concentration.

**Targeted analysis of methionine cycle intermediates.** Methionine cycle intermediates were identified and quantified using a high-resolution accurate mass (HRAM) mass spectrometer (Q-Exactive Plus, ThermoFisher Scientific GmbH, Bremen, Germany) coupled with an UHPLC system (Vanquish, ThermoFisher Scientific, Bremen, Germany). Analytes were separated using a X Select HSS T3 XP column, 100 Å, 2.5 µm, 2.1 mm × 100 mm (Waters), using a binary system A water with 0.1% formic acid, B: acetonitrile with 0.1 formic acid with a flow rate of 0.1 mL/min and the column temperature was kept at 30 °C. Gradient elution was conducted for untargeted metabolomics analysis. Methionine cycle intermediates were identified using a Targeted-SIM (t-SIM) with a resolution of 70,000, 5e$^4$ AGC target, 200 ms injection time, and 1.0 $m/z$ isolation window. The following ions were quantified: Methionine –> 149.0.5084, S-adenosyl methionine –> 398.13724, homocysteine –> 135.03540, S-adenosyl-homocysteine –> 385.12800. Quantitation was performed using Trace finder 4.1, using genesis detection algorithm, nearest RT, S/N threshold 8, min peak height (S/N) equal to 3, peak S/N cutoff 2.00, valley rise 2%, valley S/N 1.10. The relative response for each methionine intermediate was calculated by dividing the peak area of the analyte to the internal standard peak area and further normalized to protein concentration. L-methionine (2,3,3,4,4-D5; methyl-D3, 98%) (Cambridge isotopes DLM-6797-PK) and S-(5′-Adenosyl)-L-methionine-(S-methyl-$^{13}$C) (Sigma 798231) were used at a final concentration of 100 nM as internal standards for the analysis.

**Inhibition of 5MTHF uptake by folic acid.** Synchronized worms were grown on 10 cm NGM plates (three to four plate, ≈8000 worms) containing different concentrations of folic acid (0, 10, 100, 500, 1000, 2500 µM) until they reach the young adult stage. A single pulse of isotopic labeled 5MTHF-glutamic acid $^{13}$C$^{15}$N at a concentration of 5 µM was added on top of the plate. Worms were harvested after 2 h after the addition of the labeled compound. Worms from three plates (≈6000 worms) were collected and washed three times using buffer solution M9. Metabolite extraction was conducted as described above. The quantification of labeled vs unlabeled folates was performed using the previous methodology and adding the MRM transitions for $^{13}$C$^{15}$N label 5MTHF. Each experiment was repeated four times.

**Labeled folate incorporation.** To elucidate the in vivo kinetics of folic acid and 5MTHF, we used dynamic metabolite flux analysis. Synchronized worms were grown on NGM plates until they reached the young adult stage. A single pulse of isotopic labeled 5MTHF-glutamic acid $^{13}$C$^{15}$N at a concentration of 5 µM or folic acid-glutamic acid-$^{13}$C$_5$,$^{15}$N in a concentration of 10 nM was added on top of the plate. RNAi treatment for $dhfr$-1 knockdown was performed prior to the pulse. For the folic acid-glutamic acid-$^{13}$C$_5$,$^{15}$N incorporation experiment (Supplementary Fig. 3), worms were harvested 1 or 2 h after the addition of the labeled compound. For the kinetics experiment (Fig. 2e), worms were collected for metabolite extraction at seven time points (0, 10, 30, 60, 120, 240, 360 min). Worms from three plates were collected and washed three times using M9 buffer solution. Metabolite extraction was conducted as described above. Quantification of labeled vs unlabeled folates was performed using the previous methodology, adding the MRM transitions for $^{13}$C$^{15}$N label 5MTHF and folic acid-(glutamic acid-$^{13}$C$_5$,$^{15}$N). Each experiment was repeated four times.

**Statistical data.** GraphPad Prism Version 8.1 software was used for graphics and statistical testing. Metabolomics data sets were analyzed using the Fisher test and Hochberg–Benjamin false discovery rate test. Individual metabolites and targeted metabolomics were analyzed using one-way ANOVA and Dunnett's correction test. Life span experiments were analyzed using the log-rank Mantel–Cox test.

**Reporting summary.** Further information on research design is available in the Nature Research Reporting Summary linked to this article.

## Data availability
The following databases were used: HMDB (v4.0) (https://hmdb.ca/), KEGG pathway 2020 (https://www.genome.jp/kegg/pathway.html), Chemspider (http://www.

chemspider.com/), METLIN 2008 (http://metlin.scripps.edu/). The data that support the findings of this study are available within the paper (and supplementary information files) or from the corresponding author upon reasonable request. Source data are provided with this paper.

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

## Acknowledgements

We thank Dr. Orsolya Symmons for the critical reading of the manuscript and the MPG for research support. We additionally thank the bioinformatics core facility for assistance (MPI-AGE).

## Author contributions

A. Antebi and A. Annibal designed experiments and wrote the paper. A. Annibal carried out all experiments. R.G.T. performed *isp-1* supplemented with 5MTHF life span and *isp-1* qPCR. M.F.S. performed flux analysis experiments. H.T. performed Q35 and Q40 experiments and *daf-2(e1370) dhfr-1i* life span. M.M.K.A. prepared mouse samples and extracted metabolites from tissues. M.F.S., H.T., and C.L. gave technical assistance. S.G. and L.P. generated the transgenic mouse line and provided the tissue sample for the targeted metabolomic analysis.

## Funding

## Competing interests

The authors declare no competing interests.

## Ethics declarations

Mouse tissues were kindly provided by Linda Partridge. This study was performed in strict accordance with the recommendations and guidelines of the Federation of European Laboratory Animal Science Associations (FELASA). The protocol was approved by the "Landesamt fuer Natur, Umwelt und Verbraucherschutz Nordrhein-Westfalen".
