## [Peer Review File · Nature Communications]

Reviewers' Comments:

Reviewer #1:

Remarks to the Author:

This study shows that there is a common metabolic signature for three well-described and independent longevity pathways in *C.elegans*: a mutation that disrupt insulin signalling (*daf-2*), a mutation that mimic dietary restriction (*eat-2*) and a mutation that slows down mitochondrial respiration (*isp-1*). All these interventions cause alterations in the metabolism of folate which in turn causes methionine restriction. With the exception of *eat-2* mutants, the other longevity pathways are not exposed to dietary restrictions, yet metabolically they seem to operate as if they were.

This article is important and should be published in a high-visibility journal such as *Nature Communications* for three reasons. First, Methionine restriction is one of the few dietary manipulations known to robustly extend healthspan in mammals and therefore understanding the core mechanisms in model organisms is likely to have immediate clinical applications. Second, to understand longevity, we need to better understand metabolism and the metabolome is not necessarily well correlated with transcription. Therefore, phenotyping long-lived animals using metabolomic tools is key to the advancement of the field. This work presents a rare and very well performed study where the main assays are metabolomic-based. The clever use of genetics and metabolomics paves the road and raises the standard for future studies in the field. Finally, this study presents robust evidence to assert that one-carbon metabolism is a causal mechanism of longevity.

However, the article requires three main areas of revision. First, the description of the methodology is not complete and requires careful revision. Second, some key controls are missing and third, the study should be completed by looking at known readouts and molecular downstream mechanisms such as mitophagy and autophagy. In addition, the lifespan extension caused by alterations in one-carbon metabolism are small compared to the longevity of *daf-2*, *isp-1* and *eat-2* mutants. The discussion needs to address the partial nature of their findings as one yet only limited conserved pro-longevity module. I also found that the tables are too much to be followed easily and if the editors allow, I would suggest that the most important data is presented as word-tables, where the statistics can be easily followed.

Reviewer #2:

Remarks to the Author:

The manuscript by Annibal et al. reports the exciting and largely unexpected finding that in *C. elegans* several, seemingly unrelated longevity pathways converge on the one-carbon folate cycle. Specifically, it is shown that longevity associated with perturbation of insulin signaling, disruption of the mitochondrial electron transport chain, or reduced food intake confer longevity in part because of downregulation of the folate cycle, which results in reduced SAM/methionine production. This is a significant finding of broad interest given the conserved nature of these longevity pathways and of the underlying connection of folate metabolism with methionine/SAM production.

While the main conclusions are well supported by extensive data, from a combination of metabolomics, RNAi, and mutant profiling (but see comment on methionine supplementation below), the manuscript in its current form falls short of connecting the main finding – folate cycle as the target of THE most important aging pathways, likely via methionine restriction – to what is already known about the effects of methionine restriction and targeted direct perturbation of methionine or SAM generation. For example, it is well established that methionine restriction improves insulin sensitivity and longevity in mice (see e.g. <https://www.ncbi.nlm.nih.gov/pmc/articles/PMC4207389/> and references 1-6 therein), *Drosophila* and other organisms. Precedent demonstrating the importance of the folate cycle and methionine for longevity should be discussed and contextualized extensively in the Discussion section, which in its current form reads a bit like a “laundry list” of other, likely unrelated findings from the metabolomics part of this study.

With regard to the role of methionine, the Discussion remains surprisingly tentative, given the strength of the evidence (and the statement at the end of the abstract that methionine restriction is the common output). The sentence (line 366) "An important unresolved question is what is the mechanism by which dhfr-1i and 367 5MTHF reduction trigger longevity?" is confusing, given that all evidence points toward methionine restriction, as then discussed (though lacking depth) in the following paragraph.

Lastly, I am wondering why the authors did not test whether methionine or SAM supplementation can also rescue phenotypes that are due to 5MTHF depletion (or are rescued by 5MTHF supplementation). Such experiments would seem straightforward and could help delineate to what extent longevity due to folate cycle regulation is essentially due to restriction of endogenous methionine production.

Other points:

line 115: It should be clarified which m/z values the software associates with folate metabolism. Then, standards should be used to confirm the identities of the putative folate-related metabolites. Perhaps this has been done, but the tables lack this information.

line 149: "We observed that FA, THF,4-fold". Here, actual data must be presented, not just average and p-values. This should be done in the form of bar graphs, with individual data points shown, as per Nature Comm. author instructions.

line 239: Is this a 1.5-2-fold average increase across the set of genes? How does that compare to the changes observed for metr-1I?

Minor comments:

line 74: "were changed in the each of " remove "the". There are many similar error/mistakes, and the entire text and Methods section should be proof read carefully.

line 90: "Uracil, isoleucine, glutamine, lysine, appeared 90 higher in all genotypes (isoleucine), but..." "(isoleucine)" seems out of place.

line 101: extraneous "(".

line 110: I believe this should be Supplementary Table 4. The numbering of the Tables seems confused, please double check the numbering and file names.

line 114 ""branched chain" instead of "branch chain"

line 162: "extraneous "significantly"

line 739: "Metabolites concentrations" – should be "Metabolite concentrations"

Reviewer #3:

Remarks to the Author:

Annibal et al here present a comparative metabolomics analysis of long-lived *C. elegans* mutants that leads them to identify reduced 5MTHF and increased THF as a pro-longevity metabolic signature conserved all the way to mammals.

The presented metabolomics datasets are of unique value. It is the first time that measurements of the metabolic dysregulation associated to the major longevity pathways is performed in parallel in the same conditions, and hence data can be confidently compared. The authors also demonstrate that supplementation of folate intermediates normally enriched in the long-lived mutants extends *C. elegans* lifespan, and that supplementation with the intermediates normally depleted in the long-lived mutants rescues their longevity phenotype. Further, the proaging intermediate (5MTHF) is reduced, and the pro-longevity intermediate (THF) is increased in the brain and liver of long-lived *irs*^{-/-} mice. Hence, the folate-mediated regulation of lifespan described here will open new avenues of investigation in animal longevity.

Nevertheless, at this stage, the manuscript has relevant weaknesses that need to be addressed (see detailed list below). More importantly, it remains to be defined, or at least hypothesized, how 5MTHF could specifically accelerate and THF specifically delay aging. As of now, the discussion summarizes the current knowledge on the link between folate metabolism and aging. However, no mechanism by which 5MTHF and THF would specifically affect aging and proteostasis in the mutants tested here is proposed.

Therefore, the work has great potential but improvements are needed.

Major points:

1. List the provider, catalog number, and the amount of internal standard per mg of dried tissue used in the untargeted metabolomics of worms, and the targeted metabolomics of worms and mice.

Also, what is the internal standard used to normalize methionine intermediates?

2. What is the criteria applied to represent metabolites in Fig. 1C? For instance, acetoacetic acid is not in the list of measured metabolites (main figure or supplementary tables). On the other hand, valine is in the list; however, it is elevated in *eat-2*, reduced in *daf-2*, and does not change in *glp-1*.

3. Line 812: Were the presented results obtained from 5 independent biological replicates (different dates, plates, worm egg preps, etc) or otherwise? Lines 818 & 837: Were the internal standards added together with the chloroform:methanol mix or not?

4. Extended Fig. 1A.

The metabolic network used to overlay the metabolic modules found to be changing in the longevity mutants corresponds to a plant. This reviewer recommends the authors instead use as scaffold the *C. elegans* metabolic reconstructions published by the Kaleta or Walhout labs. Also, label each red dot with a number and provide list of gene or metabolite names as a separate table.

5. To facilitate the comparison between longevity pathways, organize the metabolic pathways displayed in Extended data Fig 1c-f in alphabetical order and use some color-coding to distinguish shared from uniquely enriched pathways.

6. Line 77 indicates that tyrosine metabolism is enriched in *glp-1* animals. However, tyrosine metabolism is not displayed in Extended data Fig. 1c, In addition, although the heat map in Fig. 1a shows increased abundance of tyrosine in *glp-1* animals, also *daf-2* animals show increased abundance of this amino acid; however, tyrosine is not mentioned as enriched in *daf-2* and also not presented in the enriched pathways in Extended data Fig. 1d. Similarly, serine metabolism is noted in line 81 as enriched in *eat-2*; however, serine metabolism is not displayed in Extended data Fig. 1e.

These examples illustrate how the displayed data and the conclusions in the text either do not align, or more information is needed to allow the reader to follow the criteria used by the authors to choose the which pathways are displayed and/or highlighted in the text.

7. Describe PIUmet input and output parameters such as threshold for inclusion, % of identified versus unknown metabolites after running dataset, etc.

8. Which was the denominator to define metabolite/pathway enrichment (e.g. the totality of peaks -identified plus unknown- that the author's found? The totality of *C. elegans* metabolites predicted by WormJam)?

9. If the isotopic labeling of worms shows that incorporation of 5MTHF plateaus as early as 120 min, which was the criteria used to measure FA and 5MTHF in *E. coli* after a 24h incubation?

10. Inaccurate references to data make the manuscript at times unclear.

For instance, the metabolites mentioned between lines 89 and 95 are followed by a reference to Extended Data Fig. 2. However, the mentioned figure does not show the expected data. Instead, Extended Data Fig. 2 shows the metabolites significantly changing in 3 out of 4 genotypes; hence, the figure reference in line 95 is incorrect.

Similarly, line 110 refers to Supplementary table 3. However, the only table named "Table 3" this reviewer was able to find corresponds to a list of qRT-PCR primers. Please revise all the references.

11. Fig. 1C aims to illustrate the metabolic pathways preferentially dysregulated in the longevity mutants. However, the references correlating size of the circles with level of enrichment are

missing. Therefore, although mentioned in line 116, it is not possible to independently conclude that some metabolites/enzymes are "highly enriched". Importantly, p-values (or other statistical measure) for the enrichment, and the denominator used to estimate the enrichment are not presented.

12. This reviewer recommends moving the 2nd panel of Extended data Fig. 2 (FA/methionine metabolism) to the main body of the paper. Maybe making a Fig.2 that contains these data plus main figure panel 1D?

13. If the authors model implies that what matters is the relative FA/5MTHF abundance, then it would be important to present that ratio for WT, long-lived mutant worms, and worms supplemented with pro-longevity and pro-aging metabolites. Currently, the data are presented for the individual metabolites and relative to WT. Hence, it is not possible to define the relative abundance of FA to 5MTHF.

14. Include the controls WT + FA and dhfr-1i + FA in Fig. 2b.

15. Given the role of DHFR-1 in the conversion of FA into DHF, and DHF into THF, it would be important to differentiate whether FA or DHF (or both) act as pro-longevity metabolites. Is there a technical limitation to measure the levels of DHF? Does DHF supplementation alter WT or dhfr-1i lifespan?

16. Didn't the authors find changes in cdc-42 levels between WT and some of the long-lived mutants? How were the data treated to account for differences in primer efficiency?

17. Do dhfr-1 or mthf-1 expression levels change in animals supplemented with FA or 5MTHF?

18. How do the authors interpret that WT animals supplemented with 5MTHF not living shorter than unsupplemented controls (Fig. 2b) if according to Fig. 2C the supplementation leads to a ~2 fold increase in THF and a 1.5 fold increase in 5MTHF and MN?

19. Include missing condition 'WT dhfr-1i + FA' in Fig. 2C.

20.

If this reviewer is accurately interpreting the data, comparing the levels of folate compounds to the aging phenotypes shows that ML and FO (but not FA, THF, 5MTHF and MN) correlated with the aging phenotype of *C. elegans*.

Can the authors elaborate on this difference between the data and the interpretation of the results?

21. As the authors describe in the discussion, 1C metabolism is critical to the synthesis and metabolism of multiple metabolites. For instance, the authors observe nucleoside levels consistently changing in the long-lived mutants. Further, metabolites known to affect lifespan such as alpha-ketoglutarate and NMN are common among the long-lived genotypes. Hence, what was the criteria to focus the second half of the manuscript on methionine metabolism? Further, methionine metabolism does not seem to be a salient metabolic signature in the long-lived mutants.

22. Define how the *C. elegans* orthologues/homologs of the mammalian genes known to be responsive to low methionine levels were selected. Add a supplementary table with the list of mammalian genes, E-value of homolog *C. elegans* selected, and cutoff used to not test all homolog genes.

23. More important, although the impact of dhfr-1 on methionine metabolism is an interesting observation, it does not help to understand how the abundance of FA and 5MTHF affect lifespan

and proteostasis. Therefore, instead of expanding, the authors may need to more deeply explore the mechanism by which FA and 5MTHF regulate lifespan and age-related deterioration, or at least delineate in the discussion testable hypotheses specifically linking 5MTHF to accelerated aging, and THF to delayed aging.

24. Can the data presented in Fig. 4e be interpreted as daf-16 partially suppressing dhfr-1-mediated longevity, suggesting a concerted action between daf-16 and another transcription factor?

25. Does mutation of daf-16 abrogate the induction of the methionine metabolism genes tested in Fig. 3d?

26. Line 278-9: The authors state: "Given that dhfr-1 is a regulatory target of these pathways"; however, a supporting reference is lacking.

27. Lines 300-301: Authors state "Folic acid and THF were increased up to 2-fold in both brain and liver in Irs1-/-." However, in Fig. 5c the levels of FA do not change significantly.

28. Lines 319-320: The authors state: "...we discovered that folate cycle intermediates and their enzymes represent a convergent focal point for longevity across conserved signaling pathways.". This reviewer agrees with the first part of this conclusion because supplementation of folate cycle intermediates modestly rescues daf-2 and isp-1 longevity. However, the only enzyme tested (dhfr-1) does not alter the lifespan of the long-lived mutants. Therefore, the second half of the conclusion is not supported by the data.

Statistics:

1. Show individual data points in all bar graphs and box plots.
2. Provide details on the parameters used for PCA analysis displayed in Extended Data Fig. 1b
3. Provide details on the parameters used for enrichment analyses displayed in Fig. 1c and Extended Data Fig. 1c-f, including denominator used to calculate enrichment.
4. Define the whiskers in all figures using box plots (e.g. min to max, or 10-90 percentile, etc)
5. Define error bars in figures 2d, 2e, 3d, 4c, 4d, and 4f.
6. Add mean and SD to figures 2f & g.
7. Were the 5 mice used in Figure 5 part of the same litter and/or grown in parallel, or these are completely independent replicates. Clarify these points in the methods or legend.

Minor points:

1. Typo in 2 F & G: 5MTF should be 5MTHF
2. Panels in Extended Fig. 2 are vertically overstretched.
3. Subtitle "Metabolites changed in common across pathways" may read more easily as:
Changing metabolites common across pathways
OR
Metabolites changed across pathways
4. Line 90: Uracil, isoleucine, glutamine, lysine, appeared higher in all genotypes (isoleucine),... Authors may want to delete or revise isoleucine between parentheses.
5. Line 44: Rephrase the sentence to make clear that the pathways are not "reduced insulin/IGF and mTOR signaling..." but that instead: Reduced flux through the xx, xx, and xx pathways remodels metabolism, proteostasis, and immunity towards extended survival

6. Line 86: Please replace "four pathways" with "four longevity mutants"
7. "Folic acid and 5MTHF acid were added in aqueous solution into the NGM agar at the indicated concentrations of 10 nM". Does this imply that the final concentration in the plate is 10nM, or that 10nM was the concentration of the aqueous solution added to the plates?
8. Typo in line 162: significantly is written twice.
9. Some lifespan tables denote pvalues with commas (as in 0,0001)
10. Clearly denote which lifespan replicate corresponds to the curves shown as main figure panels.
11. Fig. 4f can be moved to Extended data
12. Typo in line 725: genotype is daf-2 (not daf-1).
13. Typo in line 731: Mantel-Cox was likely used in panels a & h.
14. Typo in line 784: change for data "were" plotted.
15. Lines 190 and 812: write estimated number/worms processed per sample instead of number of plates.
16. Line 375: The authors state: "or indirectly through general control and mTOR signaling". It is unclear what "indirectly through general control" means.
17. Given that the data are available it would be a great additional resource if the authors would compare the abundance and types of metabolites differentially enriched or reduced in WT cultured at 25C relative to 20C.

REVIEWER COMMENTS

Reviewer #1

This study shows that there is a common metabolic signature for three well-described and independent longevity pathways in *C.elegans*: a mutation that disrupt insulin signaling (*daf-2*), a mutation that mimic dietary restriction (*eat-2*) and a mutation that slows down mitochondrial respiration (*isp-1*). All these interventions cause alterations in the metabolism of folate, which in turn causes methionine restriction. With the exception of *eat-2* mutants, the other longevity pathways are not exposed to dietary restrictions, yet metabolically they seem to operate as if they were.

This article is important and should be published in a high-visibility journal such as Nature Communications for three reasons. First, Methionine restriction is one of the few dietary manipulations known to robustly extend healthspan in mammals and therefore understanding the core mechanisms in model organisms is likely to have immediate clinical applications. Second, to understand longevity, we need to better understand metabolism and the metabolome is not necessarily well correlated with transcription. Therefore, phenotyping long-lived animals using metabolomic tools is key to the advancement of the field. This work presents a rare and very well performed study where the main assays are metabolomic-based. The clever use of genetics and metabolomics paves the road and raises the standard for future studies in the field. Finally, this study presents robust evidence to assert that one-carbon metabolism is a causal mechanism of longevity.

We thank the reviewer for their strong words of support!

However, the article requires three main areas of revision. First, the description of the methodology is not complete and requires careful revision. Second, some key controls are missing and third, the study should be completed by looking at known readouts and molecular downstream mechanisms such as mitophagy and autophagy. In addition, the lifespan extension caused by alterations in one-carbon metabolism are small compared to the longevity of *daf-2*, *isp-1* and *eat-2* mutants. The discussion needs to address the partial nature of their findings as one yet only limited conserved pro-longevity module. I also found that the tables are too much to be followed easily and if the editors allow, I would suggest that the most important data is presented as word-tables, where the statistics can be easily followed.

Major points

This article is in itself a great resource of ageing-metabolomics. I will ask the authors to provide very detailed annotations of the methodology used, both in the laboratory and data analysis. In general, the emerging field of worm metabolomics requires standardization of methodologies and transparency of results. I have summarized my requests in the two sections below.

Questions to be clarified regarding the growth conditions of worms:

1. At what exact timing where animals collected. Young adulthood is a broad timing, please provide hours post hatching or hours post L4. Clarify the temperature at which *daf-2*, *eat-2* and *isp-1* mutants were grown at.

In the Methods, we state that:

All strains were grown and maintained on NGM agar seeded with *E. coli* (OP50) at 20°C except for the *glp-1(e2141)ts* strain, which underwent a thermal shift to 25°C leading to germline loss. Because of differing growth rates, worms were harvested for metabolomic analyses after 60 hours for WT and *eat-2*, 72 hours for *daf-2*, and 145 hours for *isp-1* worms to ensure similar biological age.

2. Indicate how many worms were grown per plate? The amount of food is likely to impact any metabolomics assay.

In the Methods, we state that:

Worm synchronization was achieved using thirty worms to perform an egg lay for two hours on a single plate. Five plates per genotype ($\approx 10,000$ worms) were combined to obtain one biological replicate. This procedure was repeated five times.

3. Indicate more details regarding the growth media, for how long was the bacteria grown. Also, details on the agar, addition of cholesterol to the media.

In the Methods section we include information about media:

NGM was prepared using 3g/L NaCl (Sigma S3014), 2,5 g/L Bacto peptode (BD 211820), 18 g/L Bacto Agar (BD 214010), 25 mM KPO₄, 0.005 mg/mL Cholesterol (Sigma C8667), 1mM MgSO₄ and 1mM CaCl₂.

E. coli OP50 Bacteria were grown overnight (18 hours) in LB media composed of 10g/L Bacto tryptone (Sigma 95039), 5g/L Bacto yeast extract (BD 212720) and 5g/L NaCl.

HT115 strains were diluted prior to seeding till OD 0.2.

Questions to be clarified regarding data analysis:

1. How were the metabolomics results normalized? Where samples normalized according to per-sample protein content? It was not clear from Material and methods. Were values normalized to internal standards?

We state in the Methods that:

A volume of lysate, which correspond to 150 μ g of proteins for each sample was subjected to Bligh and Dyer extraction (chloroform: methanol, 2:1) for 1 hour at 4°C.

Relative quantification was obtained by dividing the area of individual metabolite peaks to spiked internal standards (Leucine enkephaline, myristic acid and cysteamine sodium salt).

2. How was the untargeted and targeted metabolomics prevented from batch effect variability? Were all samples processed in parallel?

We measured intraday and interday variability which was below 10%. Samples were prepared at the same time.

Intraday and interday accuracy error

Compound	Intraday precision (n=5)		Inter-day precision n=5/day	
	tR (min) \pm SD(RSD%)	Content (nmol) \pm SD(RDS%)	tR (min) \pm SD(RSD%)	Content (nmol) \pm SD(RDS%)
Cysteamine monophosphate (ESI+)	2.26 \pm 0.01(0.51)	199.57 \pm 14.57(7.09)	2.26 \pm 0.01(0.63)	201.62 \pm 7.60(3.77)

Myristic acid (ESI -)	2.02±0.01(0.44)	231.80±13.87(5.98)	2.02±0.01(0.44)	180.2±29.24(10.24)
Leucine enkephaline (ESI+)	12.59±0.02(0.16)	207.22±3.40(1.64)	12.59±0.01(0.10)	197.45±6.46(3.27)
Leucine enkephaline (ESI-)	12.61±0.01(0.07)	208.72±6.88(3.30)	12.61±0.02(0.15)	198.24±8.97(4.52)

3. How was this new dataset compares to other published studies in *C. elegans*. The authors briefly mention this in the main text. But a more careful table comparing results with other similar studies may be a useful resource. For example, Fuchs et al 2010; Gao et al 2018 are not referenced. That the same mutants give different metabolic profiles in different studies would not make their results less valuable.

We added Supplemental Table 3 to the Supplemental Information, which contains a comparison with previous published metabolomic studies.

4. How were missing values treated in the datasets?

By default, missing values were replaced by 1/5 of min positive values of their corresponding variables. A total of 0 (0%) missing values were detected in the set of identified metabolites.

5. The authors present the results of a principal component analysis, which captures the dimensions that best explain the variability of the samples. From that analysis it is clear that samples did not show great amount of intra-sample variance, which indicates good data quality. The description of PCA in the main text is incomplete (line64) regarding this analysis as a quality control test explain what the grouping means. Although it separates according to genotypes, this analysis does show *daf-2* as different from *isp-1* and *eat-2* which seem to cluster together. It may useful to use a supervised analysis, such as partial least squares projection to latent structures (PLS), to identify spectral features contributing most to variation or separation are identified for further analysis. From this analysis, a class membership can be statistically inferred using 95% confidence ellipses.

We thank the authors and we agree that the PLS would be more informative, and have added PLS to Extended data Figure 1b. We now state in the results:

Partial least squares discriminant analysis (PLS-DA) of the biological replicates revealed a clear grouping according to genotype (Extended Data Fig. 1b), showing the high quality of the samples. In addition, both *glp-1* and WT grown at 25°C as well as *eat-2* separated from the main cluster of *isp-1*, *daf-2*, and WT grown at 20°C.

6. How were the modules presented in Figure 1A obtained?

Metabolites were manually grouped in the main categories, now indicated in the figure.

Are these significant changes?

No, the levels of all identified metabolites are represented in the heat maps, both significant and non-significant. We think it is important to show the full set of data as a

resource because trends that are non-significant in a whole metabolomic analysis might be significant in targeted metabolomics.

What was the criteria for manually curating the categories?

We manually divided the metabolites into categories such as, amino acids, glutathione metabolism, nucleotides etc. for simplicity, which we now notate in Figure 1a. Notably, it is difficult to simply categorize metabolites according to KEGG pathway/modules since single metabolites often belong to multiple biological pathways.

In order to reduce bias and avoid potential misunderstanding regarding the manual curation, we additionally show a heat map using hierarchical clustering for both metabolites and samples (Extended Data Fig 1c).

Why is it that from untargeted metabolomics, most significant changes apply to amino-acid and nucleotide metabolism? Is there a relationship between significant changes and abundance of metabolites?

By performing untargeted metabolomics it is not possible to identify all metabolites in one single analysis. We optimized this method in order to identify a large number of metabolites. The metabolomics data was obtained using a reverse phase column. We cannot exclude that the extraction method and the column could enrich certain subgroups of metabolites. Under enrichment analysis, we now state in the text:

A caveat of enrichment analysis is that it is biased towards metabolites that can be readily measured on our platform, and might not highlight individual metabolites that are significantly changed but whose metabolic pathways are not necessarily enriched.

There are many changes associated for example with regards to lipid metabolism (example, Gao et al 2018). Were there lipids internal standards in the experiment?

Yes we used myristic acid as an internal standard for non-polar metabolites.

The number of lipids detected on this platform were limited. We did not perform a separate lipidomics analysis. We now mention this in the discussion as one of the limitations of the study:

Additionally this study focused mainly on polar metabolites, and does not represent an extensive analysis of lipids.

What was the total number of metabolites detected?

Ca 144-145 metabolites

On the effect of Folate metabolism on lifespan

1. In the supplementation experiment, Folic acid is not found to change lifespan. This experiment however is missing a basic control, which is one where it is shown that FA can rescue a FA-deficient worm. I am not exactly sure what is the best assay to use, it could be done by phenotypic rescue. The negative results observed could be technical. Figure 2C shows that this may be the case, as supplementation of FA does not increase FA levels in worms, whereas 5MTHF does show rescue (Figure 2B) and it is measurable increased when supplemented. I am slightly concerned by strong claims on line 190 that are based on metabolic changes downstream of FA supplementation, however 5MTHF chase experiment and also the results from Figure 2f and g does strengthen their argument.

We agree with the reviewer and were also first puzzled by the observation since it went against our expectations. Nevertheless, we now have used labeled FA and found that it

does indeed get into the worm (Extended Data Fig 3a). At 1 and 2 hours of incorporation time, the concentration of labeled FA increases without affecting the overall folic acid concentration pool.

We therefore interpret the data as following: Although FA does not appreciably affect life span (Ext data Fig 3b), it impacts the levels of other FA intermediates (5MTHF, MN, FO) (Fig 2c) and stimulates proteoprotection (Fig 2fg). This simplest explanation is that FA does enter the worm to affect physiology, but the effect is limited because of feedback on uptake or enzymatic activity, hence we see limited effect on life span.

Based on Figure 2F and 2G, it may be possible that FA does not affect lifespan but improves healthspan of worms. Have thrashing assays been performed in N2/wt strains?

In addition, FA improves WT motility, see data below.

Each dot represents a single worm, n= 20. Folic acid concentration 10 nM, Thrashing was measured on day 5 and day 7, N= three biological replicates. Significance was assessed using two-sided t-test. Bar shows mean ± S.D, * p<0.5, ** p<0.01, ***p<0.001.

2. On the mini-RNAi screen shown in S3. What is the effectiveness of the RNAi treatments with regards to the expression of the genes they target. A 10-15% of lifespan

extension is not very large, but the KO may be partial. Is FA essential? Is it possible to do lifespans with *tyms-1* and *dhfr-1* mutants?

We have performed RNAi knockdown and measured transcript levels for the various genes. We achieved 20-30% mRNA expression levels with *dhfr-1i* and *tyms-1i* relative to *luci* controls, which is in line with the literature. We have repeated the life spans 3 times with all the candidates and obtained similar results (Figure below, Extended data Fig 3gh).

It is possible that further knockdown could give a stronger effect. However, null mutations of *tyms-1* or *dhfr-1* are lethal or sterile (www.wormbase.org). The small change in life span compared to the canonical longevity pathways is not uncommon, and suggests that canonical longevity pathways also deploy additional mechanisms to extend life. We now say in the results section:

Conceivably the effect of *dhfr-1i* and 5MTHF on life span is not as strong as *daf-2* or *isp-1* mutation because additional mechanisms are deployed to stimulate longevity of reduced insulin/IGF signaling and mitochondrial function.

In the Discussion we also say:

Our analysis of the different longevity pathways revealed congruent changes in other crucial metabolic pathways that could additionally contribute to life extension.

On the results related to Methionine Restriction

1. With regards to the *metr-1i* conserved transcriptional signature, please be precise about the number of the targets from the Tao et al paper were tested in worms, and how many conform to the expectation based on results from mammalian cells.

Tang *et al* found 906 genes by RNA-seq that were significantly changed upon methionine deprivation, and validated a subset of these genes by qPCR, which we then used for our investigation for *dhfr-1i*. We now add Supplementary Table 9 in the Supplemental Information for comparison of homology values.

2. Methionine-restriction has been shown to activate autophagy flux and mitophagy (as of Plummer et al 2019). The effect on *in vivo* reporters for these clearing mechanisms should be shown.

Thank you for the suggestion. We estimated autophagic flux by measuring the amount of LGG-1::GFP puncta, which are modestly increased by *dhfr-1i* (Figure below).

Representative images and corresponding zoom in on the pharyngeal bulb. Punta were quantified in the pharyngeal bulb for each single worm. Images were taken using 63X magnification. Scale bar corresponds to 20 μm . A total of 15 worms were analyzed per biological replicate. Significance was assessed using two-sided t-test. Bar shows mean \pm S.D., * $p < 0.05$, ** $p < 0.01$, *** $p < 0.001$.

However, a full exploration of autophagy and related processes as a downstream mechanism are beyond the scope of this paper, and we feel would be better suited for follow up work. In addition, it does not add significant novelty since all major longevity pathways stimulate autophagy.

Although interesting, a connection to mitophagy would also open up a series of questions about mitochondrial function that go beyond the scope of the paper. Further, existent mitophagy markers are not very reproducible in our hands.

To address the issue of mechanism, we have rather focused on methionine restriction *per se*, and can show that methionine supplementation reverses longevity phenotypes. This shows that methionine reduction is not only associated with *dhfr-1i* longevity, but causal.

3. Ideally, a metabolic profile of *metr-1i* should be provided alongside *dhfr-1* KO. I am aware that metabolomics is prone to batch effect variability, so *dhfr-1i* and controls would have to be run alongside.

As requested, we have performed a metabolomic profile of *metr-1i* and *dhfr-1i* side by side. Importantly, we observed significant overlap in the changed metabolites, including those involved in the methionine cycle (Extended Data Fig 4a-d).

4. The results presented in extended figure 4 a and b could be cross-compared to metabolomics data from ageing WT worms from Hastings et al 2019.

We added Supplemental Table 3, which contains a comparison with previous published works, indicating where we have results that agree or disagree.

5. The results from Figure 4F are interesting because they show that although the folate cycle is involved in several longevity pathways that share *daf-16* as a common transcriptional regulator, the effect is independent of *daf-16*. An exciting possibility is that the transcriptional regulator involved is *pha-4*, which is known to be involved in autophagy regulation. This experiment alongside with autophagy markers would greatly strengthen this article.

We imagine two models. In one scenario, *dhfr-1* acts downstream of *daf-16* and *daf-2*. In this view, *dhfr-1* mRNA downregulation would be reversed by *daf-16* mutation. *dhfr-1i* downregulation by RNAi could still nevertheless extend life span in a *daf-16* background. In another scenario, *dhfr-1* acts downstream of another transcription factor, e.g. *pha-4*, since Modencode data suggest that PHA-4 could reside at the DHFR-1 promoter.

To address this question, we performed qRT-PCR of *dhfr-1* mRNA levels in the *daf-2* background in the presence of *daf-16i* (Figure 4f), *pha-4i* (Figure below), or *luci* control. However, neither *daf-16* nor *pha-4* modulated expression of *dhfr-1* in the *daf-2* background, suggesting other transcriptional mediators must be at work.

We also analyzed the promoter region of the folic acid genes to identify possible binding sites for transcription factors using Jaspar or UniProbe motifs (Table below). Nevertheless, no transcription factor that could be readily applied to C. elegans was identified.

| Potential transcription factor Jaspar | *F38B6.4* (+) | *dao-3* (-) | *mel-32* (-) | *dhfr-1* (+) | *metr-1* (+) | *mthf-1* (-) | *tyms-1* (+) | average_motifs | Present_in_targets |
|---|---|---|---|---|---|---|---|---|---|
| SOC1 | 5 | 5 | 9 | 4 | 12 | 5 | 4 | 6.285714 | 8 |
| IRF1 | 1 | | | 1 | 1 | | | 1 | 4 |
| Pou5f1::Sox2 | | 1 | 1 | | | | 1 | 1 | 4 |
| PI | | 2 | 1 | | | | | 1.5 | 3 |
| FLC | | | 1 | 1 | | | | 1 | 3 |
| br_Z4 | | | | | 1 | | | 1 | 2 |
| TLX1::NFIC | | | | 1 | | | | 1 | 2 |
| STAT1 | 1 | | | | | | | 1 | 2 |
| pan | | | 1 | | | | | 1 | 2 |
| CDX2 | | | | | | | 1 | 1 | 2 |
| FOXP1 | | 1 | | | | | | 1 | 2 |
| Rfx1 | | | | 1 | | | | 1 | 2 |
| RFX5 | | | | 1 | | | | 1 | 2 |
| EOR-1 | | | 1 | | | | | 1 | 2 |
| GEI-11 | 1 | | | | | | | 1 | 2 |

Potential transcription factor UniProt	F38B6.4 (+)	dao-3 (-)	mel-32 (-)	dhfr-1 (+)	metr-1 (+)	mthf-1 (-)	tym-1 (+)	average_ motifs	Present in targets
Sum1	4	3	7	16	5	8	13	8	8
Srf_secondary	8	4	9	2	7	8	5	6.142857	8
Sfp1	4		10	5	10	7	4	6.666667	7
Elf3_secondary	2				1	1	1	1.25	5
Tcfap2e_secondary			1	1	1			1	4
Foxl1_primary		1	1		1			1	4
Sox21_primary				2	2			2	3
Mtf1_secondary			2				1	1.5	3
Nkx2-9_3082.1		2			1			1.5	3
Sox17_primary					1		1	1	3
Foxk1_primary					1	1		1	3
Mafb_secondary	1		1					1	3
Stb3		1			1			1	3
Foxj3_primary		1						1	2
Bcl6b_primary				1				1	2
Hbp1_secondary		1						1	2
Sfpi1_secondary				1				1	2
Sox12_primary							1	1	2
Hoxd11_3873.1							1	1	2
Lhx3_3431.1							1	1	2
Lmx1b_3433.2							1	1	2
Hoxb9_3413.1							1	1	2
Lhx5_2279.1							1	1	2
Tlx2_3498.2						1		1	2
Fkh1					1			1	2

Minor points regarding plots, figures and text:

1. Can the introduction include a summary paragraph?

At the end of the introduction we now say:

Here we show that canonical longevity pathways converge on the folate 1-carbon pathway, and cause a decrease in levels of the intermediate 5 methyl tetrahydrofolate (5MTHF). Genetic manipulation of pathways enzymes and supplementation experiments suggest that reduction of specific folate intermediates promotes longevity and proteoprotection as a common conserved mechanism that acts through methionine restriction.

2. Supplemental Figure 1a. Unless the pathways are highlighted in the map, it should be enlarged and made readable or eliminated.

We made a number key enumerating the different pathways. Supplemental Table 2: List of compounds from our metabolomics analysis mapped onto KEGG metabolic pathways (Linked to Extended Data Fig 1a).

3. Supplemental Figure 1b. Make legends larger.

We now used partial least squares projection to latent structures (PLS). We used a larger font for the legends.

4. Figure 1a: I would add categories by which the metabolites were grouped. As it is, it is not very clear what are the changes and how the different longevity mutants compare to each other. I would suggest using a clustered heat map to highlight metabolites that are different.

We now added the categories as well as a clustered heat map in the Extended data Fig 1c.

5. Line 73, the total number of metabolites that are significantly changed are identified in Figure S1C-F and this should be made clear in the text.

Extended data Fig 1d-g depicts KEGG pathways significantly changed in the genotypes, not individual metabolites and is referred to in the main text.

We reported enrichment analysis values in a separate Supplemental Table 4: Enrichment analysis of metabolites in longevity mutants (linked to Extended Data Fig 1d,e,f,g).

6. Extended data Fig 2. Add to the figure legend a justification for not showing *glp-1* vs N2 at 25C. I am not sure if the data should not be analysed using a 2-way ANOVA test because all metabolites were obtained from the same experiments.

We added the *glp-1* and WT-25 data to Extended Data Fig 2.

7. Main text line 96 onwards. Clarify what the accumulation of a metabolite means in terms of the homeostasis of the system. The increased level of a metabolite may be caused by increased production and/or decreased usage. This analysis provides no information regarding the directionality of the metabolic flux.

We think this is a point of discussion and say:

One limitation of this study is that it only provides a broad snapshot of the steady state levels of the various metabolites and does not measure metabolic flux. Whether levels change because of altered synthesis or removal remains to be seen.

8. Main text, line 115. The network provided by PIUMet confirms the results shown in Ext Fig2, because it finds further evidence for BCAA and Folic Acid metabolism dysregulation. I would use this information to further strengthen the results. This paper follows the changes in Folic Acid, but BCAA is a very interesting group of metabolites that has been shown to be link to TOR signaling pathway, but could have TOR independent functions.

We did not focus on BCAA because they have been extensively studied in the context of mTOR signaling and longevity.

9. Ext Figure 2. The labels of the genotypes have been deformed, add them straight vertical and change.

Labels have been reformatted.

10. Extended Data 3 and beyond. In general, bar plots can be misleading, I think the standard is now to show the values of each biological replicate and SE or SD bars. It can be plotted as a bar, but showing individual replicate values.

Where appropriate we now use individual data points.

11. Ext Figure 3e, explain blue-yellow color codes. Add p values, if not related to the color code.

We removed the color code. P-values are in Supplemental Table 8.

12. I think the 5MTHF experiment should be explained better in the text. Was the experiment only performed once, or are the dots an average?

We elaborated on the experiment more and now say:

Labeled folate incorporation - To elucidate the *in vivo* kinetics of folic acid and 5MTHF we used dynamic metabolite flux analysis. Synchronized worms were grown on NGM plates until they reached young adult stage. A single pulse of isotopic labelled 5MTHF-glutamic acid $^{13}\text{C}^{15}\text{N}$ at a concentration of 5 μM or Folic acid-glutamic acid- $^{13}\text{C}_5,^{15}\text{N}$ in a concentration of 10nM was added on top of the plate. RNAi treatment for *dhfr-1* knockdown was performed prior to the pulse. For the folic acid glutamic acid- $^{13}\text{C}_5,^{15}\text{N}$ incorporation experiment (Extended data Fig. 3), worms were harvested 1 hours or 2 hours after the addition of the labeled compound. For the kinetics experiment (Fig 2e), worms were collected for metabolite extraction at seven time points (0, 10, 30, 60, 120, 240, 360 minutes). Worms from three plates were collected and washed three times using M9 buffer solution. Metabolite extraction was conducted as described above. Quantification of labeled vs unlabeled folates was performed using the previous methodology, adding the MRM transitions for $^{13}\text{C}^{15}\text{N}$ label 5MTHF and folic acid-(glutamic acid- $^{13}\text{C}_5,^{15}\text{N}$). Each experiment was repeated four times.

For FA inhibitions, How long were these worms exposed to the label. At what point was the FA added, was it in plates. Describe experiment better

We now write in the method section:

Inhibition of 5MTHF uptake by folic acid - Synchronized worms were grown on 10 cm NGM plates (three to four plate, $\approx 8,000$ worms) containing different concentration of folic acid (0, 10, 100, 500, 1000, 2500 μM) until they reach young adult stage. A single pulse of isotopic labelled 5MTHF-glutamic acid $^{13}\text{C}^{15}\text{N}$ at a concentration of 5 μM was added on top of the plate. Worms were harvested after 2 hours after the addition of the labeled compound. Worms from three plates ($\approx 6,000$ worms) were collected and washed three times using buffer solution M9. Metabolite extraction was conducted as described above. The quantification of labeled vs unlabeled folates was performed using the previous methodology and adding the MRM transitions for $^{13}\text{C}^{15}\text{N}$ label 5MTHF. Each experiment was repeated four times.

13. Figure 2F, are thrashing and number of aggregates obtained from several biological replicates? If statistics was performed then I assume that each dot corresponds to a biological replicate.

Each dot corresponds to the thrashing rate of an individual worm. Three biological replicates were performed with each BR n=30 or more worms, as stated in Figure 2 legend. One representative biological replicate is shown.

14. I would find easier to get in the story if there was a general introduction of Methionine restriction in the Introduction.

Because methionine restriction is a result and not a starting point, we feel it is better handled in results and discussion.

15. In the discussion, the modest effect of lifespan extension in one-carbon metabolism mutants should be discussed in the context of additional metabolomics changes observed in this study.

We now mention in the Results section:

Conceivably the effect of *dhfr-1i* and 5MTHF on life span is not as strong as *daf-2* or *isp-1* mutation because additional mechanisms are deployed to stimulate longevity of reduced insulin/IGF signaling and mitochondrial function.

16. My suggestion to the authors is to use the metabolomics data from this study in combination with transcriptomic data and study fluxes using Flux Balance analysis. They will be able to make predictions on how different pathways relate to each other. Metabolism is very complex and inter-related and modelling can really help in this representation.

This is an excellent suggestion, but feel that the transcriptomics must also be done in the same lab and preferably in parallel. We are doing this for another study, but it lies beyond the scope of this paper.

Reviewer #2 (Remarks to the Author):

The manuscript by Annibal et al. reports the exciting and largely unexpected finding that in *C. elegans* several, seemingly unrelated longevity pathways converge on the one-carbon folate cycle. Specifically, it is shown that longevity associated with perturbation of insulin signaling, disruption of the mitochondrial electron transport chain, or reduced food intake confer longevity in part because of downregulation of the folate cycle, which results in reduced SAM/methionine production. This is a significant finding of broad interest given the conserved nature of these longevity pathways and of the underlying connection of folate metabolism with methionine/SAM production.

While the main conclusions are well supported by extensive data, from a combination of metabolomics, RNAi, and mutant profiling (but see comment on methionine supplementation below), the manuscript in its current form falls short of connecting the main finding – folate cycle as the target of THE most important aging pathways, likely via methionine restriction – to what is already known about the effects of methionine restriction and targeted direct perturbation of methionine or SAM generation.

For example, it is well established that methionine restriction improves insulin sensitivity and longevity in mice (see e.g. <https://www.ncbi.nlm.nih.gov/pmc/articles/PMC4207389/> and references 1-6 therein), *Drosophila* and other organisms. Precedent demonstrating the importance of the folate cycle and methionine for longevity should be discussed and contextualized extensively in the Discussion section, which in its current form reads a bit like a “laundry list” of other, likely unrelated findings from the metabolomics part of this study.

Thank you for the excellent suggestion. We reworked the discussion on methionine restriction.

With regard to the role of methionine, the Discussion remains surprisingly tentative, given the strength of the evidence (and the statement at the end of the abstract that methionine restriction is the common output). The sentence (line 366) “An important unresolved question is what is the mechanism by which *dhfr-1i* and 367 5MTHF reduction trigger longevity?” is confusing, given that all evidence points toward methionine restriction, as then discussed (though lacking depth) in the following paragraph.

We now changed the discussion to assert more strongly the effect of folate on methionine restriction as a common mechanism. We also discuss more fully the potential physiological outputs associated with methionine restriction based on the literature.

Lastly, I am wondering why the authors did not test whether methionine or SAM supplementation can also rescue phenotypes that are due to 5MTHF depletion (or are rescued by 5MTHF supplementation). Such experiments would seem straightforward and could help delineate to what extent longevity due to folate cycle regulation is essentially due to restriction of endogenous methionine production.

As suggested by the reviewer, we performed methionine supplementation. We found that as predicted, methionine supplementation reversed the longevity of *dhfr-1i* in three independent biological replicates (Figure below). This shows that methionine restriction is causal for the longevity.

Other points:

line 115: It should be clarified which m/z values the software associates with folate metabolism. Then, standards should be used to confirm the identities of the putative folate-related metabolites. Perhaps this has been done, but the tables lack this information.

We add this information in the text as follow:

Identification and relative quantitation of folic acid intermediates were performed on a triple quadrupole mass spectrometer (QqQMS) (TSQ Altis, ThermoFisher Scientific GmbH, Bremen, Germany), as previously published by our group as well as the confirmation and validation of the identity of folates by using standards.

The method for the analysis of folic acid intermediates was previously published by our group DOI: 10.1002/jms.4337

line 149: “We observed that FA, THF,4-fold”. Here, actual data must be presented, not just average and p-values. This should be done in the form of bar graphs, with individual data points shown, as per Nature Comm. author instructions.

Throughout the manuscript, we changed bar charts into individual data points.

line 239: Is this a 1.5-2-fold average increase across the set of genes? How does that compare to the changes observed for *metr-11*?

We add now in the text:

We then characterized *dhfr-1i* knockdown, and found that it induced a similar signature, causing a significant 1.5-2-fold increase in the expression of these genes similar to *metr-1i* (Fig. 3d).

Minor comments:

line 74: “were changed in the each of ” remove “the”. There are many similar error/mistakes, and the entire text and Methods section should be proof read carefully.

Done.

line 90: “Uracil, isoleucine, glutamine, lysine, appeared 90 higher in all genotypes (isoleucine), but...” “(isoleucine)” seems out of place.

Done.

line 101: extraneous “(“.

Done.

line 110: I believe this should be Supplementary Table 4. The numbering of the Tables seems confused, please double check the numbering and file names.

Done.

line 114 “”branched chain” instead of “branch chain”

Done.

line 162: “extraneous “significantly”

Done.

line 739: “Metabolites concentrations” – should be “Metabolite concentrations”

Done.

Reviewer #3 (Remarks to the Author):

Annibal et al here present a comparative metabolomics analysis of long-lived *C. elegans* mutants that leads them to identify reduced 5MTHF and increased THF as a longevity metabolic signature conserved all the way to mammals.

The presented metabolomics datasets are of unique value. It is the first time that measurements of the metabolic dysregulation associated to the major longevity pathways is performed in parallel in the same conditions, and hence data can be confidently compared. The authors also demonstrate that supplementation of folate intermediates normally enriched in the long-lived mutants extends *C. elegans* lifespan, and that supplementation with the intermediates normally depleted in the long-lived mutants rescues their longevity phenotype. Further, the proaging intermediate (5MTHF) is reduced, and the longevity intermediate (THF) is increased in the brain and liver of long-lived *irs-/-* mice. Hence, the folate-mediated regulation of lifespan described here will open new avenues of investigation in animal longevity.

Nevertheless, at this stage, the manuscript has relevant weaknesses that need to be addressed (see detailed list below). More importantly, it remains to be defined, or at least hypothesized, how 5MTHF could specifically accelerate and THF specifically delay aging. As of now, the discussion summarizes the current knowledge on the link between folate metabolism and aging. However, no mechanism by which 5MTHF and THF would specifically affect aging and proteostasis in the mutants tested here is proposed.

Our findings reveal that lowered 5MTHF leads to methionine restriction, which has been shown to promote longevity in multiple species. Further, we show that methionine supplementation reverses longevity (see below). In the discussion we now extensively discuss possible mechanisms emanating from methionine restriction.

Therefore, the work has great potential but improvements are needed.

Major points:

1. List the provider, catalog number, and the amount of internal standard per mg of dried tissue used in the untargeted metabolomics of worms, and the targeted metabolomics of worms and mice. Also, what is the internal standard used to normalize methionine intermediates?

All catalog numbers have been added for each single internal standard used.

We lysed the worms or the tissue prior to the extraction. A volume of lysate, which corresponds to 150 ug was transferred in to a new tube. Before the addition of choloform methanol, we added the internal standards using the following concentrations: Leucine enkephaline 100 nM, mysistic acid 50 nM, cysteamine S Sulfate 100 nM.

We added to the method section that:

L-methionine (2,3,3,4,4-D5; methyl-D3, 98%) (Cambridge isotopes DLM-6797-PK) and S-(5'-Adenosyl) -L-methionine-(S-methyl-13C) (Sigma 798231) were used in the final concentration of 100nM as internal standard for the analysis.

2. What is the criteria applied to represent metabolites in Fig. 1C? For instance, acetoacetic acid is not in the list of measured metabolites (main figure or supplementary tables). On the other hand, valine is in the list; however, it is elevated in eat-2, reduced in daf-2, and does not change in glp-1.

The network based approach is not manually curated but generated by the algorithm.

We now add a new section in the Methods and new tables in the supplement.

We now write:

Integration of metabolomics features using the network-based algorithm PIUMet – the identified *m/z* values that were significantly changed in all genotype comparisons, were uploaded to PIUMet (<http://fraenkel-nsf.csbi.mit.edu/piumet2/>). We additional included 60 features, which were unidentified but significant in all genotypes (Sup. Table 5). We used the following parameters: number of trees 10, edge reliability 2, negative prize degree 0.0005 and number of repeats 50.

The Prize-Collecting Steiner Forest algorithm identifies metabolites and represents them as nodes, the higher the assignment score the bigger the node. The algorithm links these features based on high-confidence protein-protein and protein-metabolites interactions using two databases, HMDBv4.0 and Recon3D. Further details such as node frequency and node edge are reported in Sup. Table 6. The output was processed using the R package 'gplot' in order to visualize the cluster of metabolites and to highlight the connection between the predicted proteins and metabolites.

3. Line 812: Were the presented results obtained from 5 independent biological replicates (different dates, plates, worm egg preps, etc) or otherwise? Lines 818 & 837: Were the internal standards added together with the chloroform:methanol mix or not?

As stated in Figure 1 legend, these represent 5 independent biological replicates.

Internal standards were added prior to the extraction and we add this statement to the Methods section.

4. Extended Fig. 1A.

The metabolic network used to overlay the metabolic modules found to be changing in the longevity mutants corresponds to a plant. This reviewer recommends the authors instead use as scaffold the *C. elegans* metabolic reconstructions published by the Kaleta or Walhout labs. Also, label each red dot with a number and provide list of gene or metabolite names as a separate table.

We performed enrichment analysis using Metaboanaylst to infer enriched pathways.

We used the *C. elegans* KEGG metabolic pathways scaffold for the figure, adding for each dot a number and metabolite name, which are displayed in Table 6.

5. To facilitate the comparison between longevity pathways, organize the metabolic pathways displayed in Extended data Fig 1c-f in alphabetical order and use some color-coding to distinguish shared from uniquely enriched pathways.

We reported new enrichment charts using the new update to the Metaboanalyst platform that represents the P value and fold enrichment. For each comparison, we additionally report the statistics in Supplemental table 4.

6. Line 77 indicates that tyrosine metabolism is enriched in glp-1 animals. However, tyrosine metabolism is not displayed in Extended data Fig. 1c, In addition, although the heat map in Fig. 1a shows increased abundance of tyrosine in glp-1 animals, also daf-2 animals show increased abundance of this amino acid; however, tyrosine is not mentioned as enriched in daf-2 and also not presented in the enriched pathways in Extended data Fig. 1d. Similarly, serine metabolism is noted in line 81 as enriched in eat-2; however, serine metabolism is not displayed in Extended data Fig. 1e.

We are aware that enrichment analysis is limited and there may well be cases where individual metabolites are changing in different genotypes, but metabolic pathway enrichment does not reflect this.

We now say explicitly in the results section:

A caveat of enrichment analysis is that it is biased towards metabolites that can be readily measured on our platform, and might not highlight individual metabolites that are significantly changed but whose metabolic pathways are not necessarily enriched.

These examples illustrate how the displayed data and the conclusions in the text either do not align, or more information is needed to allow the reader to follow the criteria used by the authors to choose the which pathways are displayed and/or highlighted in the text.

We greatly appreciate the comments and we corrected any ambiguities or mistakes in the text.

7. Describe PIUMet input and output parameters such as threshold for inclusion, % of identified versus unknown metabolites after running dataset, etc.

We now added a new section in the methods as well as new tables in the supplement.

We added a sentence in the results section, saying:

Taking advantage of our unbiased metabolomics acquisition, we additionally retrieved sixty unassigned *m/z* values that were differentially regulated in all genotypes (Sup. Table 5). These uncharacterized *m/z* values were submitted to a pathway predictor software (PIUMet)¹². The PIUMet algorithm identified 75% of the submitted features, correctly assigned the already reported metabolites and pinpointed 10 new *m/z* features. To find the connection between the assigned features and a common pathway, the algorithm linked the identified compounds together based on high-confidence protein-protein and protein-metabolites interactions (PPI) (Sup. Table 6). The network revealed high confidence links of organic acids, branched chain amino acid and folic acid metabolism (Fig. 1c, Sup. Table 6). Given the biological importance of the folic acid pathway in human health and the large degree of differential regulation we observed in our data (Fig. 1a,c, Extended Data 2), we decided to follow up on the possible role of these metabolites in longevity.

8. Which was the denominator to define metabolite/pathway enrichment (e.g. the totality of peaks -identified plus unknown- that the author's found? The totality of *C. elegans* metabolites predicted by WormJam)?

We now write in the Extended Data Figure 1 d-g legend:

Enrichment analysis of significant metabolites (adj $p < .05$) obtained by uploading differentially regulated metabolites to MetaboAnalyst (<https://www.metaboanalyst.ca>) in the various mutants compared to WT. KEGG pathways containing 84 metabolite sets (KEGG, Oct. 2019) was selected. Detailed parameters are shown in Sup. Table 4.

9. If the isotopic labeling of worms shows that incorporation of 5MTHF plateaus as early as 120 min, which was the criteria used to measure FA and 5MTHF in *E. coli* after a 24h incubation?

We now additionally measured FA and 5MTHF incorporation into *E. coli* after 1 hour (Extended Data Fig.3ef).

Targeted analysis of folates intermediates of HT115 after 1 hour (left) and 24 hours (right) incubation with 10nM of FA or 5MTHF.

10. Inaccurate references to data make the manuscript at times unclear. For instance, the metabolites mentioned between lines 89 and 95 are followed by a reference to Extended Data Fig. 2. However, the mentioned figure does not show the expected data. Instead, Extended Data Fig. 2 shows the metabolites significantly changing in 3 out of 4 genotypes; hence, the figure reference in line 95 is incorrect. Similarly, line 110 refers to Supplementary table 3. However, the only table named "Table 3" this reviewer was able to find corresponds to a list of qRTPCR primers. Please revise all the references.

Thank you for pointing this out. We have gone through the entire paper to make sure references match.

11. Fig. 1C aims to illustrate the metabolic pathways preferentially dysregulated in the longevity mutants. However, the references correlating size of the circles with level of enrichment are missing. Therefore, although mentioned in line 116, it is not possible to independently conclude that some metabolites/enzymes are "highly enriched". Importantly, p-values (or other statistical measure) for the enrichment, and the denominator used to estimate the enrichment are not presented.

We rephrase the term 'enrichment' in the PIUMet analysis description and add more information in the methods and results sections.

12. This reviewer recommends moving the 2nd panel of Extended data Fig. 2 (FA/methionine metabolism) to the main body of the paper. Maybe making a Fig.2 that contains these data plus main figure panel 1D?

The 2nd panel of Extended data Fig. 2 shows untargeted metabolomics analysis of longevity pathways.

We have now performed targeted metabolomics of methionine cycle intermediates in *daf-2*, which is incorporated into Fig. 5a to keep the narrative straightforward and logical,

13. If the authors model implies that what matters is the relative FA/5MTHF abundance, then it would be important to present that ratio for WT, long-lived mutant worms, and worms supplemented with prolongevity and proaging metabolites. Currently, the data are presented for the individual metabolites and relative to WT. Hence, it is not possible to define the relative abundance of FA to 5MTHF.

In the Tables below, we provide the ratio of FA/5MTHF.

Average peak area					
	WT	daf-2(e1370)	isp-1(qm150)	eat-2(ad465)	glp-1(e214ts)
FA	25576.25	101428.2	46956.03	56701.93	24670.9
5MTHF	3250	1350.499	1700.979	1050.251	3786.507
FA/5MTHF	7.86	75.10	27.60	53.98	6.51

Average peak area						
	luci	luci + FA	luci + 5MTHF	dhfr-1i	dhfr-1i + FA	dhfr-1i + 5MTHF
FA	26476.25	25473.25	27350.75	53794.5	54870.5	49282
5MTHF	3266.75	2321.5	4938.75	1425.75	1939.25	3132.25
FA/5MTHF	8.10	10.97	5.53	37.73	28.29	15.73

Indeed, in longevity backgrounds and upon *dhfr-1i* the ratio of FA/5MTHF is increased. However, we prefer to show separate values for FA and 5MTHF across the genotypes and conditions for clarity, as it is uncertain if the ratio has more biological meaning than the individual metabolites.

14. Include the controls WT + FA and *dhfr-1i* + FA in Fig. 2b.

The controls are now added to the figure.

15. Given the role of DHFR-1 in the conversion of FA into DHF, and DHF into THF, it would be important to differentiate whether FA or DHF (or both) act as prolongevity metabolites. Is there a technical limitation to measure the levels of DHF? Does DHF supplementation alter WT of *dhfr-1i* lifespan?

We supplemented *luci* and *dhfr-1i* with 10nM DHF supplementation and found no effect on longevity (see below).

16. Didn't the authors find changes in *cdc-42* levels between WT and some of the long-lived mutants? How were the data treated to account for differences in primer efficiency?

For all qPCR quantifications, we used *cdc-42*, *act-1* and *amu-1* for normalization. We did not observe significant changes. All qPCR data are now represented as individual points.

17. Do *dhfr-1* or *methf-1* expression levels change in animals supplemented with FA or 5MTHF?

Yes as already extensively shown by Ortbauer M *et al.*, Folate deficiency and over-supplementation causes impaired folate metabolism: Regulation and adaptation mechanisms in *Caenorhabditis elegans*. *Mol Nutr Food Res.* 2016 Apr;60(4):949-56. PMID: 27061234 DOI: 10.1002/mnfr.201500819

18. How do the authors interpret that WT animals supplemented with 5MTHF not living shorter than unsupplemented controls (Fig. 2b) if according to Fig. 2C the supplementation leads to a ~2 fold increase in THF and a 1.5 fold increase in 5MTHF and MN?

It could be that some other processes become rate limiting in the wild type background, which is not uncommon in the ageing literature.

For example, epistatic regulators of life span to abolish longevity, but in some cases have relatively little effect on WT (e.g. *skn-1*, *hlh-30*, *mml-1*, *ncl-1*, and *daf-16* in some publications).

19. Include missing condition 'WT *dhfr-1i* + FA' in Fig. 2C.

The controls are now included in the figure.

20. If this reviewer is accurately interpreting the data, comparing the levels of folate compounds to the aging phenotypes shows that ML and FO (but not FA, THF, 5MTHF and MN) correlated with the aging phenotype of *C. elegans*. Can the authors elaborate on this difference between the data and the interpretation of the results?

Our data suggest that FA, ML, FO correlate with longevity, while THF, 5MTHF, MN inversely correlate with longevity.

Whether these other compounds impact life span we don't know. Not all are commercially available, and many are unstable.

Whether compounds show up or downregulation depends on the flux through the pathway (production and elimination rates, etc.). What is interesting is that 5MTHF restores the balance to all of the measured intermediates (except FA) upon supplementation of *dhfr-1i*, showing the interconnectedness of the pathway.

21. As the authors describe in the discussion, 1C metabolism is critical to the synthesis and metabolism of multiple metabolites. For instance, the authors observe nucleoside levels consistently changing in the long-lived mutants. Further, metabolites known to affect lifespan such as alpha-ketoglutarate and NMN are common among the long-lived genotypes. Hence, what was the criteria to focus the second half of the manuscript on methionine metabolism? Further, methionine metabolism does not seem to be a salient metabolic signature in the long-lived mutants.

Methionine cycle is proximal to 5MTHF and could be readily connected. The connection of folate to aKG and NMN is not as obvious and more challenging to link. Furthermore, aKG and NMN have been studied extensively. Thus we saved this for future work. We performed targeted metabolomics to examine methionine levels in *metr-1i* and observed similar metabolic regulation as *dhfr-1i*. Additionally methionine supplementation abrogates lifespan extension by *dhfr-1i*, showing that methionine restriction is causal to the *dhfr-1i* longevity. Further, as we mention in the discussion:

Notably we observed reduced levels of SAM in all four longevity pathways including *glp-1*, as well as *irs1-/-* knockout mice. Hence, methionine restriction might actually reflect s-adenosylmethionine restriction, and this node may represent an exciting entry point for all the major pathways.

22. Define how the *C. elegans* orthologues/homologs of the mammalian genes known to be responsive to low methionine levels were selected. Add a supplementary table with the list of mammalian genes, E-value of homolog *C. elegans* selected, and cutoff used to not test all homolog genes.

In the paper from Tang *et al*, 906 genes were found to be significantly changed upon methionine deprivation. We did not test all for transcriptional regulation as this could be part of future work. Tang validated by qPCR a subset of these genes, which we used for our investigation for *dhfr-1i*. We now added a supplementary table for comparison with homology values (Supplemental Table 9).

23. More important, although the impact of *dhfr-1* on methionine metabolism is an interesting observation, it does not help to understand how the abundance of FA and 5MTHF affect lifespan and proteostasis. Therefore, instead of expanding, the authors may need to more deeply explore the mechanism by which FA and 5MTHF regulate lifespan and age-related deterioration, or at least delineate in the discussion testable hypotheses specifically linking 5MTHF to accelerated aging, and THF to delayed aging. Our newly generated data reveals that *dhfr-1i* lifespan extension acts through methionine restriction through three lines of evidence. First we find that *dhfr-1i* downregulates methionine cycle intermediates, including methionine and s-adenosylmethionine, in a manner resembling knockdown of methionine synthase *metr-*

1. Second, *dhfr-1* and *metr-1* share a common transcriptional signature. Finally we have shown that methionine supplementation clearly restores normal life span to *dhfr-1i* in a dose dependent manner.

We now emphasize these points in the Discussion, and highlight the various ways in which methionine restriction could impact longevity.

24. Can the data presented in Fig. 4e be interpreted as *daf-16* partially suppressing *dhfr-1*-mediated longevity, suggesting a concerted action between *daf-16* and another transcription factor?

This seems unlikely since the percentage increase conferred by *dhfr-1i* is about the same in WT and *daf-16* (ca. 20%, Sup Table 3).

25. Does mutation of *daf-16* abrogate the induction of the methionine metabolism genes tested in Fig. 3d?

DAF-16 has little or no effect on the methionine metabolism genes in WT or *daf-2* (Figure 5b).

26. Line 278-9: The authors state: "Given that *dhfr-1* is a regulatory target of these pathways"; however, a supporting reference is lacking.

We were referring to our own data and now reference the panel.

27. Lines 300-301: Authors state "Folic acid and THF were increased up to 2-fold in both brain and liver in *Irs1-/-*." However, in Fig. 5c the levels of FA do not change significantly.

Thank you for pointing this out. We rewrote the section to clearly distinguish between significant changes and trends.

28. Lines 319-320: The authors state: "...we discovered that folate cycle intermediates and their enzymes represent a convergent focal point for longevity across conserved signaling pathways.". This reviewer agrees with the first part of this conclusion because supplementation of folate cycle intermediates modestly rescues *daf-2* and *isp-1* longevity. However, the only enzyme tested (*dhfr-1*) does not alter the lifespan of the long-lived mutants. Therefore, the second half of the conclusion is not supported by the data.

We showed that *dhfr-1i* extends life span in the WT background. If it works in a parallel pathway to *daf-2*, then *dhfr-1i* should further extend *daf-2* life span. If it works in the same pathway, then it should not further extend *daf-2* longevity. We observed the latter, and therefore conclude they work in the same pathway. The regulation of *dhfr-1* mRNA levels by *daf-2* and *isp-1* mutation gives further evidence that they act in the same pathway.

We now say;

By generating the metabolic profiles of several different long-lived mutants in *C. elegans*, we discovered that folate cycle intermediates represent a convergent focal point for longevity regulation across conserved signaling pathways.

Statistics:

1. Show individual data points in all bar graphs and box plots.

All bar charts are converted to individual data points.

2. Provide details on the parameters used for PCA analysis displayed in Extended Data Fig. 1b

Following the suggestion of another Reviewer, we now show PLS. The parameters are now added to the figure legend.

3. Provide details on the parameters used for enrichment analyses displayed in Fig. 1c and Extended Data Fig. 1c-f, including denominator used to calculate enrichment.

We now using Metaboanalyst for the enrichment analysis. Analysis parameters are added to the figure legend and in Supplemental Table 4.

4. Define the whiskers in all figures using box plots (e.g. min to max, or 10-90 percentile, etc)

We changed whiskers to individual data dots.

5. Define error bars in figures 2d, 2e, 3d, 4c, 4d, and 4f.

We clarified this aspect in the figure legends.

6. Add mean and SD to figures 2f & g.

We added the mean and SD.

7. Were the 5 mice used in Figure 5 part of the same litter and/or grown in parallel, or these are completely independent replicates. Clarify these points in the methods or legend.

We clarified in the text that

N=5 mice, each dot represent a single animal.

Minor points:

1. Typo in 2 F & G: 5MTF should be 5MTHF

We corrected it.

2. Panels in Extended Fig. 2 are vertically overstretched.

We corrected it.

3. Subtitle "Metabolites changed in common across pathways" may read more easily as: Changing metabolites common across pathways

We now say: Changed metabolites common across pathways

4. Line 90: Uracil, isoleucine, glutamine, lysine, appeared higher in all genotypes (isoleucine),...

Authors may want to delete or revise isoleucine between parentheses.

We corrected it.

5. Line 44: Rephrase the sentence to make clear that the pathways are not “reduced insulin/IGF and mTOR signaling...” but that instead: Reduced flux through the xx, xx, and xx pathways remodels metabolism, proteostasis, and immunity towards extended survival

We now say:

Downregulation of insulin/IGF and mTOR signaling, reduced mitochondrial respiration, dietary restriction and reduced hormonal signals from the reproductive system can remodel metabolism, proteostasis, stress pathways and immunity towards extended survival and longevity.

6. Line 86: Please replace “four pathways” with “four longevity mutants”

We corrected it.

7. “Folic acid and 5MTHF acid were added in aqueous solution into the NGM agar at the indicated concentrations of 10 nM”. Does this imply that the final concentration in the plate is 10nM, or that 10nM was the concentration of the aqueous solution added to the plates?

This refers to the final concentration in the plate.

8. Typo in line 162: significantly is written twice.

Removed.

9. Some lifespan tables denote pvalues with commas (as in 0,0001)

We corrected it.

10. Clearly denote which lifespan replicate corresponds to the curves shown as main figure panels.

We corrected it and added this information in the Supplemental Table 8.

11. Fig. 4f can be moved to Extended data

We corrected it.

12. Typo in line 725: genotype is daf-2 (not daf-1).

We corrected it.

13. Typo in line 731: Mantel-Cox was likely used in panels a & h.

Done.

14. Typo in line 784: change for data “were” plotted.

Done.

15. Lines 190 and 812: write estimated number/worms processed per sample instead of number of plates.

We now added the number of plates with estimation of worm number in the method section for each experiment.

16. Line 375: The authors state: “or indirectly through general control and mTOR signaling”. It is unclear what “indirectly through general control” means.

We mean that the amino acid can be a rate limiting metabolite for which it is a substrate, e.g. protein synthesis, or signal through amino acid sensing in mTOR and general control pathways

We have now largely rewritten the section on methionine restriction.

17. Given that the data are available it would be a great additional resource if the authors would compare the abundance and types of metabolites differentially enriched or reduced in WT cultured at 25C relative to 20C.

The data are in the paper, but we do not comment on it since most changes are small.

Reviewers' Comments:

Reviewer #1:

Remarks to the Author:

I think that this article is one of a kind, the authors managed to study metabolomics mechanistically, going beyond what most published studies accomplish. I am satisfied with the careful revisions that have been done in response to my review as well as in response to other reviewers.

Reviewer #2:

Remarks to the Author:

The revised manuscript by Annibal et al. addresses most of my comments and suggestions, and the writing overall seems much improved. I do have a few remaining suggestions I'd like the authors to consider:

- The methionine concentrations (20 – 40 millimolar) are much, much higher than the 5MTHF levels (10 nanomolar) required to have a lifespan effect. Since there is no reason to believe that methionine is not effectively taken up (e.g. since labeling with D3-methionine works very well in worms) I wonder whether there could be other methylation products, more directly accessible through 5MTHF than via methionine, that underlie the lifespan effects. Obviously, this speculative question can't be expected to get resolved in the present manuscript, but the authors could comment on this curious difference (1,000,000-fold) in required concentrations.
- The authors should better address mine and the other reviewers' previous comment that the effects of one-carbon metabolism are not nearly as strong as those of e.g. daf-2 mutation. The sentence that was added in the Results section ("Conceivably the effect...") feels awkward and really this should be dealt with in the Discussion section.

Minor comments:

- line 232: punctuation after "length"
- line 308: it would make sense to include metr-1 here, and also to already refer to it when discussing the folate pathway.
- line 471: "2 amino" should be "2-amino"
- line 783: "represents"
- "Sup. Table 1" and similar – I think Nature style may be "Supplemental ..."
- "20mM", "10nM", "1mM", etc.: space needed between numeral and unit.
- line 1139: "Folic" should be "folic"

Reviewer #3:

Remarks to the Author:

Annibal's et al manuscript has been greatly improved and I am satisfied with the answers to my previous concerns. At this point, only a few text and content changes are still needed.

1) Include raw values of each of the repeats in Table S1

2) Add to discussion a hypothesis about how even though 1C metabolism is significantly altered, some its most directly impacted amino acids (glycine and serine) are not.

3) Add a supplementary figure with a graph representing the biochemical pathways described in lines 152 – 175.

4)

Lane 115 reads: "Because glp-1 mutants lack germline and often showed dissimilar metabolic features, we limited our search to the metabolites that were commonly and significantly...." However, all the tested mutants show metabolic alterations (e.g. CR mutant eat-2). Therefore, please replace the sentence with something like: glp-1 animals are sterile, which may explain their more discrepant metabolic profile. Hence, we focused on the study of the non-sterile genotypes.

5)

Lane 446 reads: "Methionine metabolism ramifies into multiple intermediates and pathways associated with longevity. Hence, methionine restriction might actually reflect s-adenosylmethionine restriction, and this node may represent an exciting entry point for all the major pathways."

However, since SAM but not 5MTHF are reduced in glp-1 and methionine levels are actually increased in this germline mutant, 5MTHF and/or methionine restriction may not be necessary to reduce SAM, and different metabolic changes may lead to SAM depletion in different longevity models. Consequently, this complexity needs to be discussed and the implicit notion that the 5MTHF-methionine-SAM link is conserved across longevity models and phylogeny needs to be tone down.

REVIEWERS' COMMENTS

Reviewer #1 (Remarks to the Author):

I think that this article is one of a kind, the authors managed to study metabolomics mechanistically, going beyond what most published studies accomplish. I am satisfied with the careful revisions that have been done in response to my review as well as in response to other reviewers.

We thank the reviewer for his/her kind words.

Reviewer #2 (Remarks to the Author):

The revised manuscript by Annibal et al. addresses most of my comments and suggestions, and the writing overall seems much improved. I do have a few remaining suggestions I'd like the authors to consider:

- The methionine concentrations (20 – 40 millimolar) are much, much higher than the 5MTHF levels (10 nanomolar) required to have a lifespan effect. Since there is no reason to believe that methionine is not effectively taken up (e.g. since labeling with D3-methionine works very well in worms) I wonder whether there could be other methylation products, more directly accessible through 5MTHF than via methionine, that underlie the lifespan effects. Obviously, this speculative question can't be expected to get resolved in the present manuscript, but the authors could comment on this curious difference (1,000,000-fold) in required concentrations.

We thank the reviewer for this comment. We write in the discussion section:

Altogether these findings indicate that methionine restriction is causally linked to dhfr-1i life extension. Strikingly we saw similar changes in folate and methionine pools in tissues of Irs1-/- knockout mice and daf-2/InsR mutant worms, revealing that the control of the folate cycle by insulin/IGF signaling is evolutionarily conserved. Whether these changes are causally connected to life span regulation in mammals remains to be seen.

Though the metabolic link between 5MTHF and methionine seems clear, it should be pointed out that 10 nM 5MTHF and 40 mM methionine were required for dhfr-1i rescue, showing large differences in concentration. These concentration levels are consistent with the literature (Giese G. et al, eLife, 2020 supplemented worms with 25 mM methionine and Edwards C. et al, BMC Genetics, 2015), and probably reflect differences in the uptake or utilization of these compounds, but also raise the possibility that other 5MTHF/methionine derived metabolites or substrates could be involved. It also seems likely that additional amino acids other than methionine regulate longevity of dhfr-1i, namely glycine and serine, which act in the folate cycle. In this analysis, however, we were not able to measure levels of these amino acids, and thus cannot exclude their contribution in our proposed model.

- The authors should better address mine and the other reviewers' previous comment that the effects of one-carbon metabolism are not nearly as strong as those of e.g. daf-2 mutation. The sentence that was added in the Results section ("Conceivably the effect...") feels awkward and really this should be dealt with in the Discussion section.

In the discussion, we add the following sentence:

Conceivably, some of these metabolic modules could further regulate longevity of reduced insulin/IGF signaling and mitochondrial function, since daf-2 or isp-1 mutants had stronger effects on life span than dhfr-1i, and 5MTHF supplementation only partially reduced longevity of these two strains.

Minor comments:

- line 232: punctuation after “length”

Done.

- line 308: it would make sense to include metr-1 here, and also to already refer to it when discussing the folate pathway.

We thank the reviewer for the suggestion. However, we prefer to keep separate the two section and describe first the RNAi lifespan mini screening and then add the metr-1 in the methionine restriction paragraph.

- line 471: “2 amino” should be “2-amino”

Done.

- line 783: “represents”

Corrected.

- “Sup. Table 1” and similar – I think Nature style may be “Supplemental ...”

Done. We now change to “Supplementary”.

- “20mM”, “10nM”, “1mM”, etc.: space needed between numeral and unit.

Done.

- line 1139: “Folic” should be “folic”

Done.

Reviewer #3 (Remarks to the Author):

Annibal’s et al manuscript has been greatly improved and I am satisfied with the answers to my previous concerns. At this point, only a few text and content changes are still needed.

1) Include raw values of each of the repeats in Table S1.

We added the raw (non-normalized data for each repeat in the table S1).

2) Add to discussion a hypothesis about how even though 1C metabolism is significantly altered, some its most directly impacted amino acids (glycine and serine) are not.

We now add to the discussion:

Though the metabolic link between 5MTHF and methionine seems clear, it should be pointed out that 10 nM 5MTHF and 40 mM methionine were required for dhfr-1i rescue, showing large differences in concentration. These concentration levels are consistent with the literature 43,44,

and probably reflect differences in the uptake or utilization of these compounds, but also raise the possibility that other 5MTHF/methionine derived metabolites or substrates could be involved. It also seems likely that additional amino acids other than methionine regulate longevity of dhfr-1i, namely glycine and serine, which act in the folate cycle¹⁶. In this analysis, however, we were not able to measure levels of these amino acids, and thus cannot exclude their contribution in our proposed model.

3) Add a supplementary figure with a graph representing the biochemical pathways described in lines 152 – 175.

The biochemical pathway is already described in the Figure 1d.

4)

Lane 115 reads: “Because glp-1 mutants lack germline and often showed dissimilar metabolic features, we limited our search to the metabolites that were commonly and significantly....” However, all the tested mutants show metabolic alterations (e.g. CR mutant eat-2). Therefore, please replace the sentence with something like: glp-1 animals are sterile, which may explain their more discrepant metabolic profile. Hence, we focused on the study of the non-sterile genotypes.

We now write in the result section:

glp-1 mutants are sterile and lack germline, which could give rise to a disparate metabolic profile. We therefore limited our search to the metabolites that were commonly and significantly regulated in three non-sterile genotypes, daf-2, eat-2 and isp-1 (adj p<.05, Extended data Fig. 2, Supplementary Table 1).

5)

Lane 446 reads: “Methionine metabolism ramifies into multiple intermediates and pathways associated with longevity. Hence, methionine restriction might actually reflect s-adenosylmethionine restriction, and this node may represent an exciting entry point for all the major pathways.”

However, since SAM but not 5MTHF are reduced in glp-1 and methionine levels are actually increased in this germline mutant, 5MTHF and/or methionine restriction may not be necessary to reduce SAM, and different metabolic changes may lead to SAM depletion in different longevity models. Consequently, this complexity needs to be discussed and the implicit notion that the 5MTHF-methionine-SAM link is conserved across longevity models and phylogeny needs to be tone down.

We now write in the discussion section:

SAM metabolism ramifies into multiple intermediates and pathways associated with longevity. Notably, we observed reduced levels of SAM in four longevity pathways including glp-1, as well as Irs1-/- knockout mice (though levels of SAM are likely controlled through different routes in glp-1 given the differences in 5MTHF regulation).